# ScenarioNet: Open-Source Platform for Large-Scale Traffic Scenario Simulation and Modeling

**Quanyi Li[‡*], Zhenghao Peng[§*], Lan Feng[†*],**
**Zhizheng Liu[†], Chenda Duan[§], Wenjie Mo[§], Bolei Zhou[§]**
[‡]University of Edinburgh, [†]ETH Zurich, [§]University of California, Los Angeles

## Abstract

Large-scale driving datasets such as Waymo Open Dataset and nuScenes substantially accelerate autonomous driving research, especially for perception tasks such as 3D detection and trajectory forecasting. Since the driving logs in these datasets contain HD maps and detailed object annotations that accurately reflect the real-world complexity of traffic behaviors, we can harvest a massive number of complex traffic scenarios and recreate their digital twins in simulation. Compared to the hand-crafted scenarios often used in existing simulators, data-driven scenarios collected from the real world can facilitate many research opportunities in machine learning and autonomous driving. In this work, we present *ScenarioNet*, an open-source platform for large-scale traffic scenario modeling and simulation. ScenarioNet defines a unified scenario description format and collects a large-scale repository of real-world traffic scenarios from the heterogeneous data in various driving datasets including Waymo, nuScenes, Lyft L5, Argoverse, and nuPlan datasets. These scenarios can be further replayed and interacted with in multiple views from Bird-Eye-View layout to realistic 3D rendering in MetaDrive simulator. This provides a benchmark for evaluating the safety of autonomous driving stacks in simulation before their real-world deployment. We further demonstrate the strengths of ScenarioNet on large-scale scenario generation, imitation learning, and reinforcement learning in both single-agent and multi-agent settings. Code, demo videos, and website are available at https://metadriverse.github.io/scenarionet.

## 1 Introduction

Autonomous Driving (AD) is revolutionizing mobility with benefits like reliable transportation, travel comfort, and fuel efficiency. As a safety-critical application, it is important to evaluate the AD stack thoroughly and ensure its reliability and safety before the real-world deployment [58]. Unlike the standardized evaluation of the perception module of AD stack on large-scale real-world annotated data [45], the planning and control modules are often evaluated in simulators with synthetic hand-crafted scenarios [21, 51, 27, 56]. These testing scenarios, often overly simplified, cannot reflect the complexity of real-world conditions like map structures and diverse traffic participant behaviors [7]. Moreover, it is time-consuming and requires domain knowledge to craft such testing scenarios, especially those with many traffic participants [22]. Thus, it remains challenging to scale up the evaluation and the training of the AD's decision-making in a wide range of situations.

Collecting traffic scenarios from real-world driving data can address the data accessibility issue. By replaying the real-world traffic scenarios in simulation, we can further benchmark the decision-making of the AD system and improve the generalizability of the data-driven planning and control components. However, several critical issues need to be addressed when building a large-scale

---

[*] Quanyi Li, Zhenghao Peng and Lan Feng contribute equally to this work.

37th Conference on Neural Information Processing Systems (NeurIPS 2023) Track on Datasets and Benchmarks.

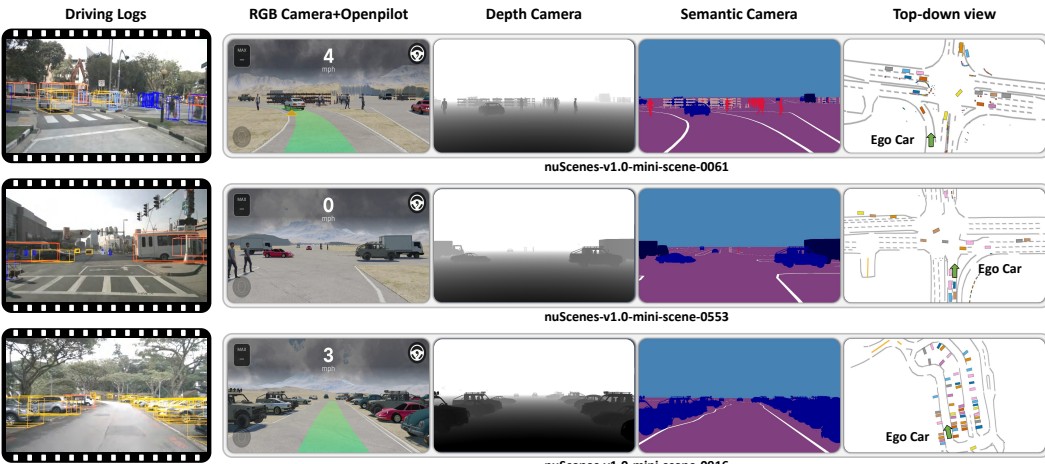

| Driving Logs | RGB Camera+Openpilot | Depth Camera | Semantic Camera | Top-down view |
| --- | --- | --- | --- | --- |

nuScenes-v1.0-mini-scene-0061

nuScenes-v1.0-mini-scene-0553

nuScenes-v1.0-mini-scene-0916

Figure 1: Snapshots of three scenarios extracted from *nuScenes* and their corresponding interactive environments with multiple views and sensors including RGB camera, depth camera, and semantic camera. The RGB sensor can be used for end-to-end driving systems like Openpilot [10].

data-driven simulation platform: First, annotated driving data is precious and we would like to use as much data as possible. However, it is difficult to achieve this since publicly available driving data are released by different organizations and in various formats. This creates a bottleneck in aggregating the data volume from multiple sources for training and evaluating AD systems. Second, existing datasets are closely coupled with specific simulators or toolkits built for ad hoc applications, which largely limits the possibility of data sharing. For example, nuPlan [6] with 1500+ hours driving data aims at single-agent Imitation Learning (IL); Nocturne [54] with 500+ hours Waymo motion data is mainly designed for multi-agent Reinforcement Learning (RL) under partial observation; SimNet built on 1000 hours L5 motion dataset targets at scenario generation. Researchers who want to use nuPlan data for multi-agent RL have to spend a significant effort on building a new simulator or bridging Nocturne [54] and nuPlan dataset. This greatly restricts the potential of cross-dataset training. Furthermore, driving datasets like nuPlan [6], nuScenes [5], and Argoverse [8, 59] provide raw sensor data like images and point clouds, but existing 2D data-driven simulators can not make use of them. The lack of z-axis and 3D graphics prevents training visuomotor policies and evaluating end-to-end AD stacks, like Openpilot [10] in simulation.

To tackle the aforementioned challenges, we introduce *ScenarioNet*, an open-source platform to unify the heterogeneous driving data from various sources for large-scale traffic scenario simulation and modeling. It extracts a massive number of traffic scenarios in a unified format from various datasets, including Waymo motion datset [18], nuScenes dataset [5], L5 motion prediction dataset [26], Argoverse dataset [8, 59], and nuPlan dataset [6]. These scenarios can be replayed and reacted through the MetaDrive simulator [30] in multiple views from BEV layout to realistic 3D rendering. *ScenarioNet* enables many machine learning applications like large-scale scenario generation, imitation learning, and reinforcement learning in both single-agent and multi-agent settings. With the built-in ROS bridge, the digital twins of the traffic scenarios can further serve as testbeds for AD stacks, where in our case we evaluate the open-source one like Openpilot [10]. We conduct a range of experiments based on *ScenarioNet*. First, we train a large scenario generation model on a joint dataset and analyze the embedded space of traffic scenarios to identify the relationship across various datasets. After that, we conduct cross-dataset single-agent training and testing experiments to show the advantage of real-world data for training ML-based planners. In addition, multi-agent RL/IL experiments are conducted on Waymo dataset [32, 18] to show the support for agent-based scenario generation. Lastly, we test Openpilot [10], an end-to-end AD stack in reconstructed scenarios. We hope our open-sourced *ScenarioNet* with its flexible APIs and toolkits empowers the community with many new research opportunities and propels the development of safe and generalizable AI for autonomous driving.

## 2 Related Work

**Synthetic scenario database.** The majority of the existing scenario databases are based on synthetic data which are either handcrafted [3, 62, 16, 21, 17, 29, 63], or generated based on rules [7, 30, 24, 53,

| Simulator | Supported Dataset | RL | IL | Multi-agent RL & IL | Scenario Generation | AD Stack Testing | 3D Rendering |
|---|---|---|---|---|---|---|---|
| nuPlan-devkit [49] | nP | | ✓ | | | | |
| DriverGym [28] | L5 | ✓ | ✓ | | | | |
| Nocturne [54] | W | ✓ | ✓ | ✓ | | | |
| TrafficSim [48] | - | | | | ✓ | | |
| SimNet [48] | L5 | | | | ✓ | | |
| VISTA [2, 57] | W, nS | ✓ | ✓ | ✓ | | | ✓ |
| **ScenarioNet** | W, L5, nP, nS, Ag | ✓ | ✓ | ✓ | ✓ | ✓ | ✓ |

Table 1: Comparison of representative data-driven simulators. *ScenarioNet* supports Waymo (W), L5 Lyft, nuPlan (nP), nuScenes (nS), and Argoverse (Ag) datasets and various applications and tasks.

66, 4] and adversarial attacks [13, 14, 40, 68, 67] which, in particular, aims at building safety-critical cases. Synthetic scenarios are predominantly described using OpenScenario [3, 17, 16], Scenic [21], and Python language [62, 29], allowing the definition of map structures, initial actor displacements, and subsequent behaviors. These scenario descriptions can be parsed for building scenarios in CARLA [15] seamlessly. Representative databases are OSC-ALKS [3], highway-env [29], DI-drive [16], and SafeBench [62] which mainly provides safety-critical scenarios.

**Real-world scenario database.** To the best of our knowledge, CommonRoad [1] and nuPlan [6] are the only real-world datasets curated specifically for the development and testing of the planning module. CommonRoad lacks sensor data and has a limited number of scenarios, while nuPlan [6] has more than 1500+ hours of annotated driving data. Though high-quality data is provided by nuPlan, the accompanying *nuPlan-devkit* is limited to open-loop/close-loop IL. Extra effort is needed for it to support wide applications like RL-based planner training.In this case, *ScenarioNet* can complement single-agent and multi-agent RL training with nuPlan data. Besides, *ScenarioNet* has a 3D rendering support which can additionally make use of the sensor data in nuPlan dataset for investigating end-to-end AD stack and some interdisciplinary tasks like street scene reconstruction. Furthermore, *ScenarioNet* can incorporate more motion prediction datasets like Waymo dataset [44], nuScenes dataset [5], and L5 dataset [26] for cross-dataset training and testing.

**Data-driven simulation.** We refer to driving simulators capable of replaying real-world scenarios or generating realistic traffics as data-driven simulators. In Table 1, we summarize the properties of existing data-driven simulators in terms of widely supported real-world data sources and applications, and tasks. Each simulator has a different set of supported datasets. The support for RL and IL entails single-agent driving policy learning for the ego vehicle only, while MARL (IL) needs to control additional vehicles present in the scenario. Thus scenario generation represents methods requiring not only actuating objects in a similar way to trajectory forecasting [35, 55] but composing new traffic scenarios by inserting vehicles into an empty map. TrafficGen [19] exemplifies how *ScenarioNet* enables building the training dataset and loading generated scenarios for RL training. Among all the simulators, only the simulation environment provided by *ScenarioNet* supports AD stack testing. The last column shows that *ScenarioNet* and VISTA [2, 57] are the only two platforms that utilize camera and lidar data for 3D rendering support. Though both VISTA and *ScenarioNet* can train driving policies, there is a distinct difference. VISTA achieves closed-loop training and synthesizing new sensor data by applying relative transformations to the recorded sensor data. In contrast, *ScenarioNet* reconstructs scenarios with mid-level object representations, such as vehicle positions and velocities. Despite the loss of raw sensor data fidelity, this design enables broader scope of capabilities including not only closed-loop training but also scenario-based AD stack testing and scenario generation.

## 3 System Design of ScenarioNet

Fig. 3 illustrates the system design of the proposed *ScenarioNet*. It collects a massive number of real-world traffic scenarios by converting heterogeneous driving data from different sources, like Waymo dataset [47], nuScenes/nuPlan dataset [6, 5], and so on. Meanwhile, it can replay the imported scenarios in the MetaDrive simulator [30] through the unified scenario description, supporting various

machine learning applications and tasks, such as AD stack testing, single-agent learning, multi-agent learning, and scenario generation. We describe each component of *ScenarioNet* in more detail below.

## 3.1 Unified Scenario Description

A unified scenario description format is the key for extracting scenario data from different sources. As shown in Fig. 2, we define the unified scenario description format as a nested dictionary with 4 top-level keys. Each scenario in the source dataset will be converted into `scenario_id.pkl`.

**Map Features.** Each scenario description file stores map features consisting mainly of lanes and lane lines. Both lanes and lane lines have a polyline field recording the lane centerline or lane line itself. Given an object, we can know how far it moves along the lane and the lateral distance to the lane centerline by converting its position from the global Cartesian coordinate to the lane centerline's Frenet coordinate. On the other hand, contact checks can be executed between a vehicle and a lane line to decide whether the vehicle is driving across the lane line. For lanes, it has two extra fields: polygon, and connectivity. Polygon can be used to query if a given object is on the lane, and the drivable area can be determined by combining polygons of all lanes. Connectivity records the neighbor lanes, next and previous lanes of the given lane.

**Objects.** In most raw driving data, the road objects are usually stored in a frame-centric way where the user need to visit all time frames before collecting the state sequence of an object. In *ScenarioNet*, the object representation is object-centric meaning that querying the object dictionary with an object ID will return the state sequence across the episode. This is convenient for downstream applications which are mostly object-centric. Take trajectory prediction as an example. For getting the ground truth of the motion of an object, user can query the description with `[object_id]["position"]` and `[object_id]["velocity"]`. The queries will return the whole trajectory and velocity. However, not all objects are valid throughout the whole episode and hence we introduce a `[object_id]["valid"]` key for indicating which frame the object presents. For loading the objects into the simulator, we synchronize the position, heading, velocity, and rotation rate between the logged data and

```
{"map_features": {
    "map_obj_id_1": {
        "centerline": [...],
        "type": "lane",
        "connectivity": {...},
        "polygon": [...]
    },
    "map_obj_id_2": {
        "polyline": [...],
        "type": "white_solid_line"
    },
},
"objects": {
    "object_id_1": {
        "position": [...],
        "heading": [...],
        "velocity": [...],
        "valid": [...],
        "size": (l, w, h),
        "type": "VEHICLE"
    }
},
"traffic light": {
    "light_id_1": {
        "states": [...],
        "position": (x, y, z),
        "lane": "map_obj_id_1"
    }
},
"metadata": {
    "dataset": "nuscenes",
    "time_interval": 0.1,
    "summary": {
        "num_objects": 53,
    }
}}
```

Figure 2: Example of Scenario Description

the simulated object entity when the `[object_id]["valid"]` indicator is `True` at the given frame. Otherwise, the object will disappear and be destroyed. By doing so, the digital twin scenario can be faithfully reconstructed in MetaDrive simulator.

**Traffic Lights.** Similar to objects, traffic lights are stored in a dictionary with the ID as the key and properties as the value. The light states can be accessed by `[light_id]["state"]` throughout one episode, where each element can be Red, Yellow, Green, or Unknown and represent the status of the corresponding lane. As each object can localize which lane it is on, it can easily know if there is a traffic light on the current lane or an oncoming lane.

**MetaData.** `["metadata"]` exists at each level of the nested dictionary, storing miscellaneous data. For example, for an object, the `[object_id]["metadata"]` can store the instance ID provided by the original dataset. For the top-level dictionary of scenario description, a `["metadata"]` stores the information about the source dataset, like the original scenario ID, which original file this scenario comes from, the time interval between two frames, the coordinates system of the data, *etc*. In addition, some statistics of the scenario are included such as the number of objects, the number of traffic lights, and the moving distance of each object. This information can be used to select or filter scenarios to build new databases as in Sec. 3.2.

To ensure a scenario description can be loaded into the simulator, we provide a tool script to check if necessary keys are present in the specific dictionary and if the data type of the key's value is correct. For example, `["position"]` can not be absent from the object property dictionary, and its value should be in shape $(N, 3)$ where $N$ is the episode length. As long as all necessary keys exist, users can add new fields to the scenario description to log more details about the scenario.

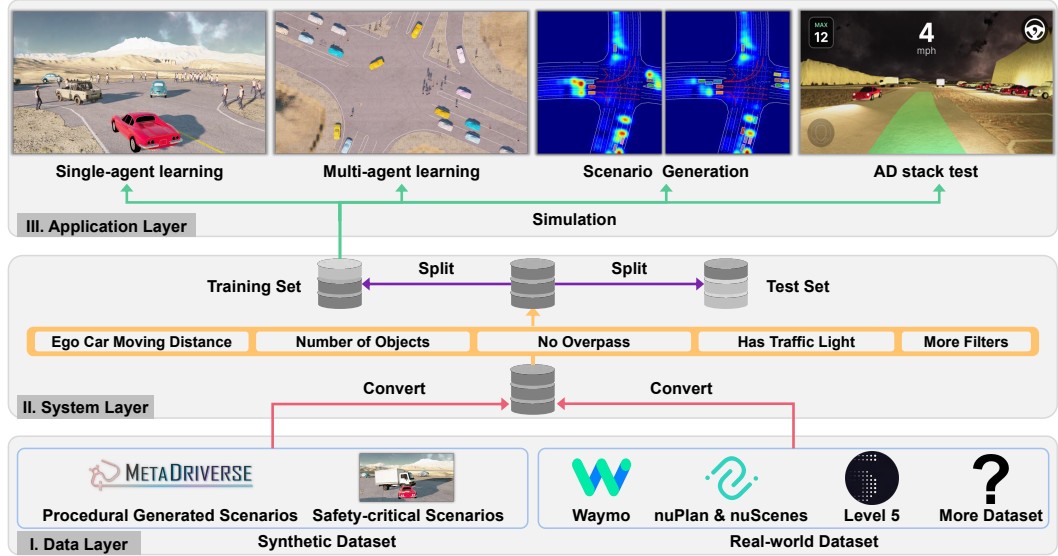

Figure 3: From bottom to top, *ScenarioNet* platform consists of the data layer, system layer, and application layer which are connected by two critical data flows, data conversion (→) and simulation (→). Data conversing unifies various data formats and stores them in an internal scenario description. The system layer then provides a set of tools to operate converted data efficiently, such as filtering (→), merging, sanity-check, splitting (→) and so on. Once the database is ready, it can be loaded into MetaDrive for large-scale simulation and various applications.

## 3.2 Scenario Database

The database in *ScenarioNet* is a folder containing two summary files `dataset_summary.pkl` and a `dataset_mapping.pkl`. `dataset_summary.pkl` stores a dictionary with keys indicating which scenarios are included in this database and values storing the metadata of each scenario. `dataset_mapping.pkl` records the relative path of the scenario file `scenario_id.pkl` from the database folder, and hence no matter where the scenario files are we can access them through the relative path recorded in `dataset_mapping.pkl`.

**Database Operations.** *ScenarioNet* provides several operations to create and maintain the databases including raw data conversion, sanity-check, merging, splitting, sampling, and filtering. A new database can be formulated with a series of operations. *Conversion* is the basic operation for creating a database by processing driving data in various formats to the unified scenario description. This operation produces a database: a folder with summary files `dataset_summary.pkl` and `dataset_mapping.pkl` and a number of converted scenario files with filenames like `sd_nuscenes_scenes-0061.pkl`. Currently, we provide data converters for parsing data from Waymo [47], nuPlan [6], nuScenes [5], L5 [26], and procedurally generated dataset [30]. It is also convenient to write new converters for users' own dataset for exploiting the power of *ScenarionNet*. After converting and building from raw data, the databases then can be operated to create new databases on demand. For example, Merging operation combines several databases together to get a larger one. Filtering operation creates a new database with scenarios satisfying specific conditions. For example, we can build a database with scenarios where ego car moving distance > 10 meters and the number of traffic participants > 200. Sanity-check is a special filter for building a new database by discarding broken scenarios that can not be loaded into the simulator. Splitting and sampling aim to divide training and non-overlapping test sets. Instead of copying the raw data when creating new dataset, we only read the summary files from source dataset and create new `dataset_summary.pkl` and `dataset_mapping.pkl`. The copy-free design and the parallel operations ensure efficiency.

**New Dataset Support.** It is easy to add support for new datasets by following these steps:

1. Download the original data and set up the official codebase for parsing this data.

2. Make a `convertor_function` which takes one scenario recorded in the original format as input and returns the scenario described in a new format. Actually, this process is like filling in the blank of the target scenario description by parsing the original data format with official APIs.
3. Scale up the data conversion with the built-in function, `write_to_directory`. It takes the `convert_function` and a list of scenario indices or original scenarios as input and writes the converted scenarios into a target directory. In this process, the multi-process parallel converting, sanity-check and other trivial steps will be finished automatically.

Though step 2 seems time-consuming, most of the work in this step is, actually, calling APIs provided by the new dataset, retrieving the desired data, and putting them in the corresponding field. A good example is `convert_waymo_scenario` where most of the subfunction is called `extract_xyz`. Since the structure and fields to fill in the scenario description are clear and automatic tools are provided, we believe it would be easy for the community to add support for more datasets.

### 3.3 Scenario Simulation

We simulate the traffic scenarios as interactive environments through an upgraded version of our MetaDrive simulator [30], which encompasses Bullet [12] physics engine, 2D Pygame rendering backend, and 3D OpenGL rendering backend. The physics engine not only simulates realistic vehicle dynamics but also efficient contact and collision detection among vehicles, objects, and map features such as lane lines. As a result, the physics simulation can run up to 500 FPS for real-world scenarios containing +100 interactive objects including pedestrians, vehicles, cyclists, and traffic barriers/cones. We use synthetic models to represent pedestrians and vehicles, so no personal information is identifiable.

**Sensors.** *ScenarioNet* supports collecting the mid-level representations of scenarios through 2D pseudo-lidar and bird's-eye view images. It also supports querying the internal simulator states for precise ground truth information about surrounding objects, such as their positions and velocities. Nevertheless, end-to-end commercial AD stacks like Openpilot [10] are emerging and a recent trend suggests it is important to consider the perception and planning system as a whole [11]. *ScenarioNet* thus provides RGB-camera, depth-camera, and point-cloud for sensor simulation. All camera outputs are synthesized via the OpenGL rendering backend, which is wrapped by the Panda3D game engine and allows users to add new visual effects or shaders via Python code easily. We optimize the camera efficiency by bridging CUDA and OpenGL so that the images rendered by OpenGL in VRAM can be converted to Pytorch [38] Tensor directly. The entire procedure occurs on the GPU, resulting in an improvement from 11 FPS to 300 FPS when simulating a 1920x1080 image. In addition, to better simulate RGB camera output, we upgrade the 3D-rendering effect of the native MetaDrive 3D renderer through deferred rendering which enables simulating photorealistic light, shadow, and material. Some examples of the supported sensors are in Fig. 1.

**Control Policies.** All objects in the scene are controlled by policies. In a single-agent setting, vehicles can be controlled by either the `ReplayPolicy` to replay logged trajectories or the `IDMPolicy` to not only replay but react to other objects. The `IDMPolicy` enables closed-loop simulation by detecting if there are objects present on its future trajectory and adjusting the target velocity to avoid collision with the leading object. In this way, when the ego car deviated from the recorded trajectory, vehicles behind it can react to this change and show yielding behavior. Also, the ego car can be controlled by the `ReplayPolicy` to the replay trajectory or the `EnvInputPolicy` to accept external control commands from `env.step()`. Moreover, by assigning the `EnvInputPolicy` to traffic vehicles, the task can be turned into a multi-agent setting. This policy-based design in *ScenarioNet* allows for mixed-policy simulation and provides high customizability. For instance, one can develop a policy that wraps a ROS bridge to connect open-source AD stacks to the simulation.

## 4 Experiments and Applications

We conduct a range of experiments and demonstrate various applications of *ScenarioNet*. We introduce how we built databases containing scenarios from different sources and then use these databases to conduct experiments on single-agent and multi-agent learning and AD stack testing.

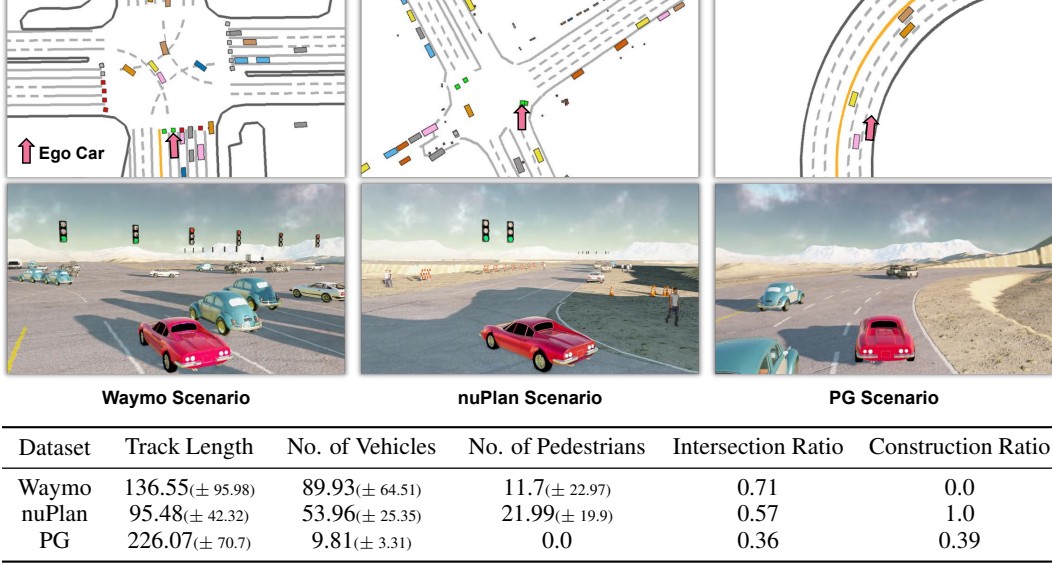

| | Waymo Scenario | | nuPlan Scenario | | PG Scenario |
|---|---|---|---|---|---|

| Dataset | Track Length | No. of Vehicles | No. of Pedestrians | Intersection Ratio | Construction Ratio |
|---|---|---|---|---|---|
| Waymo | $136.55_{(\pm 95.98)}$ | $89.93_{(\pm 64.51)}$ | $11.7_{(\pm 22.97)}$ | 0.71 | 0.0 |
| nuPlan | $95.48_{(\pm 42.32)}$ | $53.96_{(\pm 25.35)}$ | $21.99_{(\pm 19.9)}$ | 0.57 | 1.0 |
| PG | $226.07_{(\pm 70.7)}$ | $9.81_{(\pm 3.31)}$ | 0.0 | 0.36 | 0.39 |

Table 2: The statistics of the Waymo, nuPlan, and PG database.

## 4.1 Database Construction

**Waymo.** Waymo database is built with the Waymo Motion dataset [47]. The raw data contains 500+ hours of driving logs divided into 70,000+ scenarios with 20 seconds duration for each scenario. We convert all of them into internal scenario descriptions and then filter out the scenarios where the ego vehicle moves less than 10 meters or where overpasses exist.

**nuPlan.** Though nuPlan [6] has more than 1500 hours of driving data, we only use those recorded in Boston as we want to keep all databases in the experiments to have similar sizes. After filtering out scenarios whose ego car moves < 10 meters, we finally collect 50,261 20-second scenarios.

**PG.** PG stands for procedural generation which is used to generate infinite maps and traffic according to a set of pre-defined roadblocks and traffic initialization rules. It is the default scenario generation method of MetaDrive simulator [30]. We set the traffic density to 15 vehicles per 100 meters and 2 roadblocks in each scenario to keep scenario length consistent with real-world scenarios. We finally collected 50,000 scenarios to form the PG database.

Table 2 provides statistics about the three datasets. *Track length* stands for the average moving distance of the ego car across all scenarios, and PG scenarios have the longest moving distance. Waymo scenarios are relatively more complex compared to others, as they have more vehicles and more intersection scenarios, which is reflected in Fig. 4 as well. The *Construction Ratio* represents the ratio of scenarios containing traffic cones and barriers out of all scenarios. nuPlan-boston database has traffic cones and barriers in all scenarios, while no objects are contained in the Waymo database. More database construction details are in Appendix F.

## 4.2 Visualization of Scenario Embeddings with Scenario Generation Model

To reveal the differences and gaps across three databases, we use the internal embedding from a scene generation model, TrafficGen [19], to visualize the scenario snapshots from all three databases. The state initializer in TrafficGen utilizes an encoder-decoder structure that encodes the traffic scenario into a latent space and then decodes the initial states distribution of traffic participants, enabling the sampling and generation of a new scenario snapshot. We randomly select 1000 scenarios from each database, feed them into TrafficGen to obtain the embeddings of these scenarios and then employ t-SNE [52] to visualize them in Fig. 4. We can see there is a noticeable domain gap between synthetic and real-world data. Only two synthetic intersection scenarios are close to real-world scenarios and bridge the synthetic scenarios and real scenarios. Additionally, even among real-world scenarios, a minor distribution gap remains, primarily due to different road network topologies in different regions where the data is collected. **These findings suggest that solely relying on driving data from one**

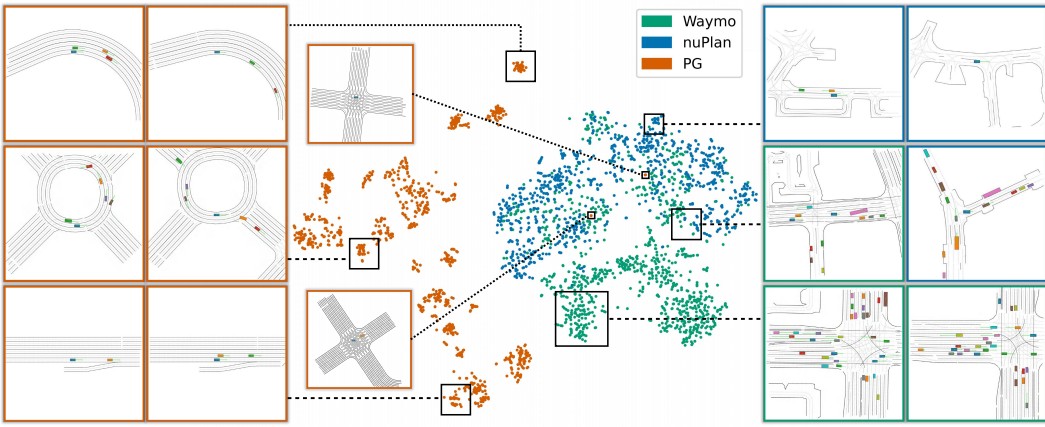

Figure 4: t-SNE visualization of the feature embeddings of scenarios from three databases. Scenarios from nuPlan and Waymo datasets are much more similar, while synthetic PG data has a clear domain gap with both real-world datasets. We also plot the representative scenarios in clusters. As shown in the left-side figures (□), synthetic scenarios can be neatly categorized into distinct clusters, while real-world scenarios shown on the right side are too complex to identify clear boundaries. Nevertheless, the right-side figures show scenarios from nuPlan (□) have simpler map structures and mild traffic, compared to the Waymo scenarios (□). In addition, there are two outliers that confuse the model, causing it to cluster them with real-world scenarios. The visualization of the two synthetic scenarios indicates that the intersection scenario can bridge the sim-to-real gap.

**region or one database is insufficient to cover all potential traffic situations.** Therefore, it is important to aggregate as many scenarios as possible. Additional showcases from different databases, visualizations of generated scenarios, and training details can be found in Appendix A.

### 4.3 Cross-dataset RL Generalization

The previous experiment reveals that the domain gap exists between synthetic scenarios and real-world scenarios. We would like to further investigate whether the domain gap affects driving policy learning. To this end, we build a data-driven RL environment where the agent should arrive at the destination of the recorded trajectories in time. Traffic vehicles behind the ego car are controlled by `IDMPolicy` to achieve closed-loop training and testing. We further split training and test sets for the nuPlan database and the PG database, respectively, and get 4 splits: *nuPlan-train*, *PG-train*, *nuPlan-test* and *PG-test*.

Each training set contains 40,000 scenarios, while each test set has 5,000 non-overlapping scenarios. The scenarios of two training splits are sorted according to the difficult score, a metric considering length and curvature of trajectory, and constitutes 100 levels for conducting the curriculum training. Each level contains 400 scenarios and a 75% *Success Rate*, the ratio of scenarios where the agent can reach the destination out of 400 scenarios, is required for the learning environment to level up. We train PPO [43] agents on *PG-train* and *nuPlan-train* splits with the curriculum training and evaluate them on *nuPlan-test* split to study whether the domain gap harms the performance for the learning-based controller. All re-

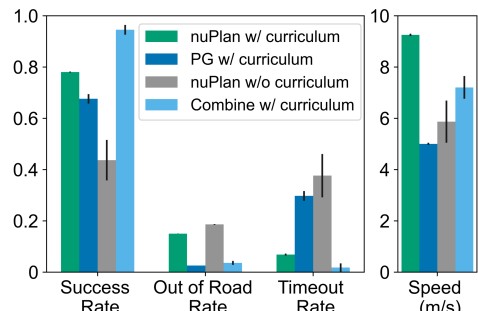

Figure 5: Evaluation on *nuPlan-test* split.

sults are averaged across 5 random seeds and the error bar shows the standard deviation. As shown in Fig. 5, when testing on held-out *nuPlan-test* set, the agents trained with *nuPlan-train* split outperform those trained with *PG-train* split in terms of *Success Rate*. This is because the agents trained in synthetic scenarios fail to learn high-speed driving skills and thus can not reach the destination in time, inducing a high *Timeout Rate*. **Thus it is necessary to train learning-based controllers with real-world data to close the sim-to-real gap**. We also ablate the curriculum training and the results

| | Route Completion | Success Rate | Reward | Cost | Average Distance | Final Distance |
|---|---|---|---|---|---|---|
| TD3 | 0.669 (±0.018) | 0.513 (±0.028) | 74.87 (±5.90) | 2.01 (±0.24) | 12.64 (±0.72) | 45.13 (±2.73) |
| PPO | 0.740 (±0.037) | 0.579 (±0.054) | 85.28 (±8.96) | 2.83 (±0.53) | 12.79 (±0.73) | 36.71 (±6.38) |
| CoPO | 0.745 (±0.024) | 0.570 (±0.043) | 85.94 (±4.60) | 2.88 (±0.32) | 12.86 (±0.63) | 34.75 (±3.78) |
| AIRL (Disc.) | 0.367 (±0.027) | 0.153 (±0.028) | - | 0.55 (±0.12) | 17.73 (±1.87) | 80.57 (±5.26) |
| AIRL (Cont.) | 0.138 (±0.016) | 0.015 (±0.009) | - | 0.42 (±0.16) | 3.63 (±1.07) | 111.23 (±7.48) |
| GAIL (Disc.) | 0.487 (±0.021) | 0.218 (±0.018) | - | 1.64 (±0.35) | 28.48 (±2.23) | 68.53 (±3.57) |
| GAIL (Cont.) | 0.421 (±0.043) | 0.189 (±0.051) | - | 1.56 (±0.52) | 27.33 (±4.72) | 75.98 (±9.02) |

Table 3: The multi-agent reinforcement learning and imitation learning results on the Waymo dataset.

suggest that **it is critical to adjust the scenario difficulty according to the agent performance in the course of training**. Finally, we build a new dataset containing 40,000 scenarios by combining half of the *PG-train* split and half of the *nuPlan-train* split. Figure. 5 shows that agents trained on these combined datasets achieve the highest *Success Rate* when evaluated on *nuPlan-test* split. This is because almost every synthetic scenario contains bends and curvated trajectories, which only exist in approximately 20% of real-world scenarios. As a result, agents acquire a better ability to follow a curved trajectory, achieving the lowest *Out of Road Rate*. **Thus aggregating synthetic scenarios to the training dataset can benefit policy learning as they improve the diversity of training data.** More details of the experiments are in Appendix B.

## 4.4 Multi-agent Cooperative Learning

*ScenarioNet* can load scenario data into an interactive multi-agent policy learning environment and generate local sensory observations for each agent. Therefore, *ScenarioNet* enables the critical research on multi-agent reinforcement learning as the agents are self-interested and the number is varying, wherein cooperative MARL methods can not be applied [39], as well as the research on multi-agent imitation learning with the challenges of learning from observation or say the action-free imitation learning, meaning that the ground-truth trajectory does not contain action but only the sequence of states (observations). *ScenarioNet* bridges the research of multi-agent decision-making with vast volume of real-world data. We load the ground-truth (GT) trajectory in the dataset and use it to form the termination condition, the reward function, and the evaluation metrics. For multi-agent RL, we use the GT trajectory to define *the displacement reward: $r_t^i =$ the displacement of agent $i$ at step $t$ projected into its underlying ground-truth trajectory*. We train independent PPO [43] and TD3 [23] agents as well as the Coordinated Policy Optimization (CoPO) agents [39]. For multi-agent IL, to address the action-free issue, we use GAIL [25] in multi-agent setting [46] but the discriminator distinguishes state-next state pair, instead of state-action pair [50]. We also train multi-agent Adversarial Inverse RL [65] (AIRL) with an additional inverse dynamics model for estimating the expert actions. The inverse dynamics model is trained concurrently with the AIRL policies and learns the action given state-next state pairs from the environment interactions. The ground-truth trajectory can be used to measure the learned behaviors, such as *the route completion rate*, the ratio between the length of projected agent trajectory and the length of GT trajectory as well as *the average and final distance* between agent trajectory with GT trajectory. The cost is the number of crashes of an agent in one episode. As shown in Table 3, we find that **the displacement reward is strong supervision to learn multi-agent behavior** as the RL can explore and exploit the reward function by interacting with the environment, without learning a discriminator as in GAIL or a reward function and an inverse dynamics model in AIRL. More details can be found in Appendix C.

## 4.5 AD Stack Testing

We released ROS bridge [33], allowing connecting *ScenarioNet* with open-sourced AD stacks like autoware [20] and planning algorithms developed by the ROS community. As a result, researchers can study self-driving systems in real-world scenarios. In our experiment, we test Openpilot [10], an end-to-end AD system taking only visual information as input. As the navigation module of Openpilot is still a beta version, the test happens in scenarios without diverged roads. As shown in Fig. 6, the Openpilot system is robust enough to operate in common scenarios like lane-keeping and stopping at traffic lights. More scenario-based test videos and details are available in Appendix E.2.

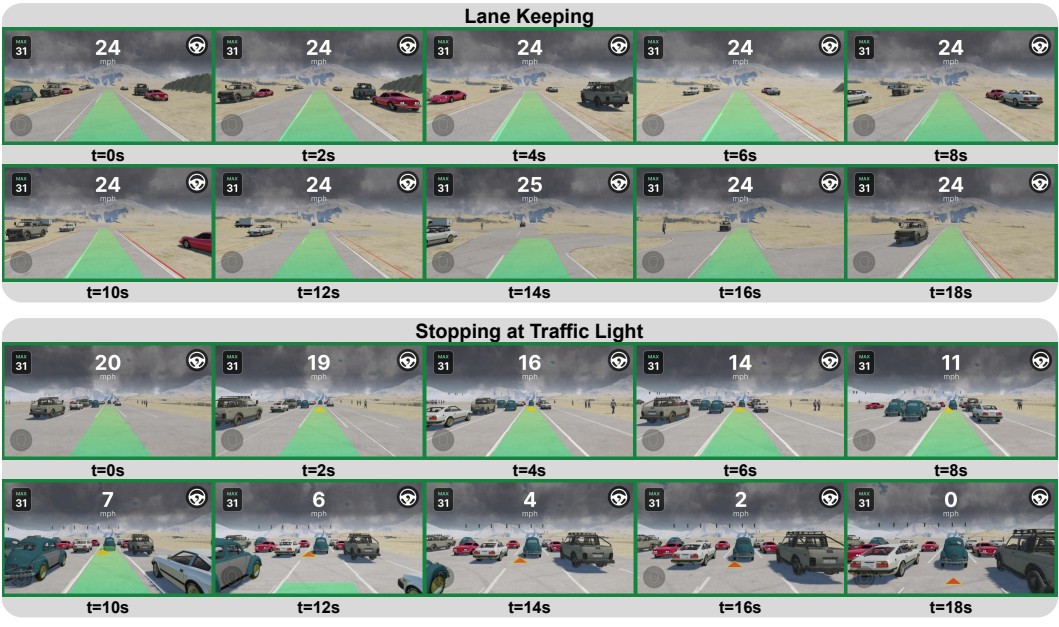

Figure 6: Openpilot can operate in real-world scenarios reconstructed with *ScenarioNet*. It can cruise on urban roads and smoothly stop at crowded intersections.

## 5   Conclusion

This work presents an open-source platform called *ScenarioNet* for simulating and modeling driving scenarios. A massive number of real-world traffic scenarios extracted from various datasets can be used for testing AD stacks, generating scenarios, and learning both single-agent and multi-agent policies. We hope it can open up many new opportunities for AI and autonomous driving communities.

**Limitation & Potential Negative Societal Impacts.** One limitation of *ScenarioNet* at the current development stage is the lack of diverse 3D assets. Currently, the assets are collected from the internet, which may produce unrealistic raw sensor data output. Future work will be to reconstruct the mesh and textures of vehicles, other objects, and background models like streets from driving logs using neural rendering. A detailed plan for improving visual fidelity is in Appendix G. In addition, several potential technical drawbacks exist in *ScenarioNet*. One drawback is that the simulation is single-thread due to the Python GIL and runs on CPU, while some latest simulators like IsaacGym [34] support GPU-based parallel simulation and thus outperform *ScenarioNet* in terms of sample efficiency. This problem can be alleviated by using multi-process tools like Ray/Rllib [37, 31] for parallel simulation. Another drawback is that *ScenarioNet* doesn't support differential simulation which allows training policy with the derivative of reward function directly [36]. In the future, *ScenarioNet* can address this by introducing a bicycle model for vehicle dynamics simulation, making the system differentiable. One potential negative societal impact is that there is no guarantee of safety for even a data-driven AD stack can pass all the testing scenarios in *ScenarioNet*. There is also a domain gap for the trained model to be deployed in the real world.

**Licenses.** *ScenarioNet* is released under Apache License 2.0. Panda3D, used in MetaDrive for 3D rendering, is under the Modified BSD license. Bullet engine is under the Zlib license. The vehicle models are collected from Sketchfab under CC BY 4.0 or CC BY-NC 4.0. For real-world data used in experiments, Waymo Open Dataset [32, 47] is under license terms available at `https://waymo.com/open`. Besides, the L5 dataset, nuPlan dataset, nuScenes dataset, and Argoverse dataset are provided free of charge under the Creative Commons Attribution-NonCommercial-ShareAlike 4.0 International Public license. The toolkits for parsing these datasets are all under Apache license version 2.0. The Openpilot system used for AD stack testing is under MIT license.

**Acknowledgement:** This work was supported by the National Science Foundation under Grant No. 2235012 and the Samsung Global Collaboration Award.

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

# Appendix

In this appendix, we provide more details about the four experiments and some scenario examples from the three databases used in the experiments. Section A is on the scenario generation model; Section B is on the single-agent cross-dataset generalization experiment; Section C is on the multi-agent reinforcement learning and imitation learning; Section E shows the interface of ROS bridge and the qualitative results of running Openpilot test. Section F shows more rendered scenario examples from each database.

Codebase, documentation, videos, and a scenario gallery are available at https://metadriverse.github.io/scenarionet.

# A Scenario Generation Model

We use all the databases (nuPlan, Waymo, PG) in scenarioNet for cross-dataset traffic scenario generation experiments. The goal is to show that scenarioNet contains diverse traffic scenario data for training deep neural networks. We conduct our experiments based on TrafficGen [19], a neural generative model previously developed for reactive traffic scenario generation.

## A.1 Task Definition

A traffic scenario is denoted as $\tau = (\mathbf{m}, \mathbf{s}_{1:T})$, which lasts $T$ time steps and contains the High-Definition (HD) road map $\mathbf{m}$ and the state series of traffic vehicles $\mathbf{s}_{1:T} = [s_1, ..., s_T]$. Each element $s_t = \{s_t^1, ...s_t^N\}$ is a set of states of $N$ traffic vehicles at time step $t$. Given an existing scenario $\tau = (\mathbf{m}, \mathbf{s}_{1:T})$, the goal of TrafficGen is to learn to generate **new** traffic scenarios $\tau' = (\mathbf{m}, \mathbf{s'}_{1:T'})$ that have similar distribution with $\tau$ and different states $\mathbf{s'}$ and longer time steps $T'$. After training, TrafficGen takes $\mathbf{m}$ as input and generates $\tau'$ as a totally different scenario.

## A.2 Model Architecture

**Traffic Scenario Encoder.** TrafficGen uses vectorization to encode map and vehicle information, representing lanes as sets of vectors, each vector representing a small region. A vector-based coordinate system is established for each small region, with each vector comprising a start point $p_i^s$, endpoint $p_i^e$, and information about vehicles in this region. Thus, a traffic snapshot $\tau$ at time step $t$ is represented as $\tau_t = \mathbf{v} = v_i{}_{i=1}^I$. Cross attention mechanism is applied to the unordered set $\mathbf{v}$ to fuse information from different regions into a single context vector.

**Decoder for Scenario Generation.** TrafficGen places vehicles by generating a set of weights for all regions, which is then turned into a categorical distribution. The local position of a tentative vehicle is modeled by a mixture of $K$ bivariate normal distributions, as are heading, speed, and size of the vehicle. Autoregressive sampling is used to create a traffic snapshot. A motion forecasting model is used as the trajectory generator.

## A.3 Experiment Setting

We train a scenario generation model TrafficGen with mixed data. Specifically, we randomly sample 30% data from nuPlan, Waymo, and PG. We filter out the scenarios with less than 8 cars and crop a rectangular area with a side length of 120m centered on the ego vehicle. Each of the 20s scenarios is split into 10 traffic snapshots with 2s intervals. The training is executed on servers with 8 x Nvidia 2080TI and 256 G memory in Distributed Data-Parallel (ddp) mode. We set the feature size to be 1024, and use 3-layer MLPs with hidden dimensions of [1024, 512, 256] for attribute modeling. The training takes 16 hours for 100 training epochs and the learning rate decays by 20% at every 30 epochs. The detailed hyperparameters are shown in Table 4.

We plot some generated traffic scenarios in Fig. 7 and the dynamics in Fig. 8. It shows that the models trained on the processed scenario data from ScenarioNet can generate realistic and diverse traffic behaviors and interactions.

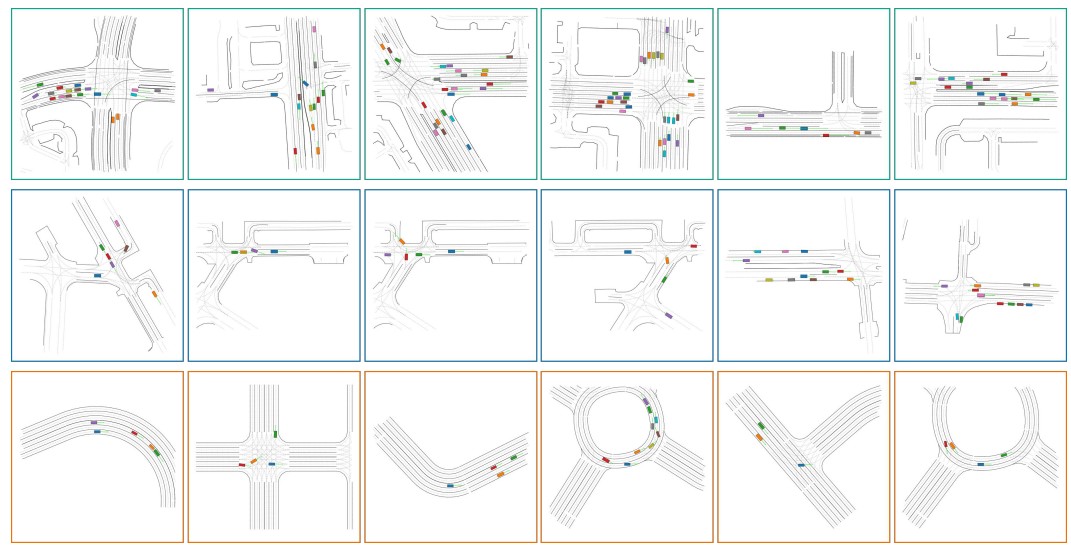

Figure 7: Traffic scenarios generated from the model trained on different databases: Waymo (□), nuPlan (□), and PG (□).

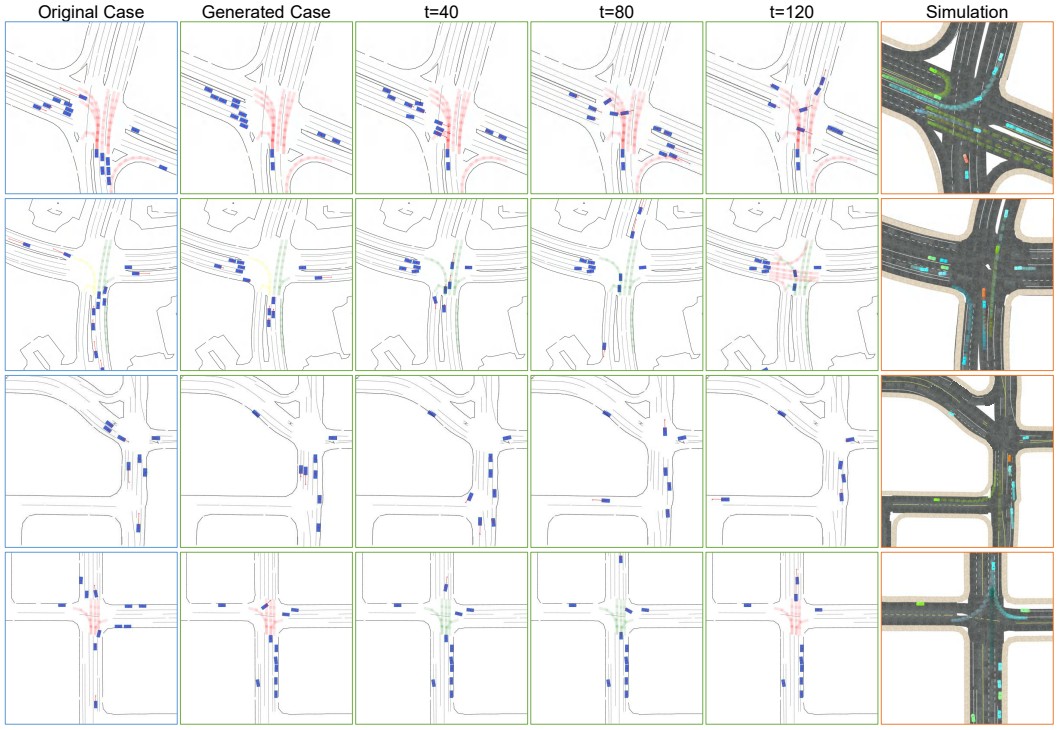

Figure 8: Dynamics of the generated traffic scenarios. The first column is the original case. The middle columns show the generated scenarios at different timesteps. The last column shows the corresponding scenarios imported in the simulation. The green and red dashed lines indicate the traffic light status of this lane at the intersection.

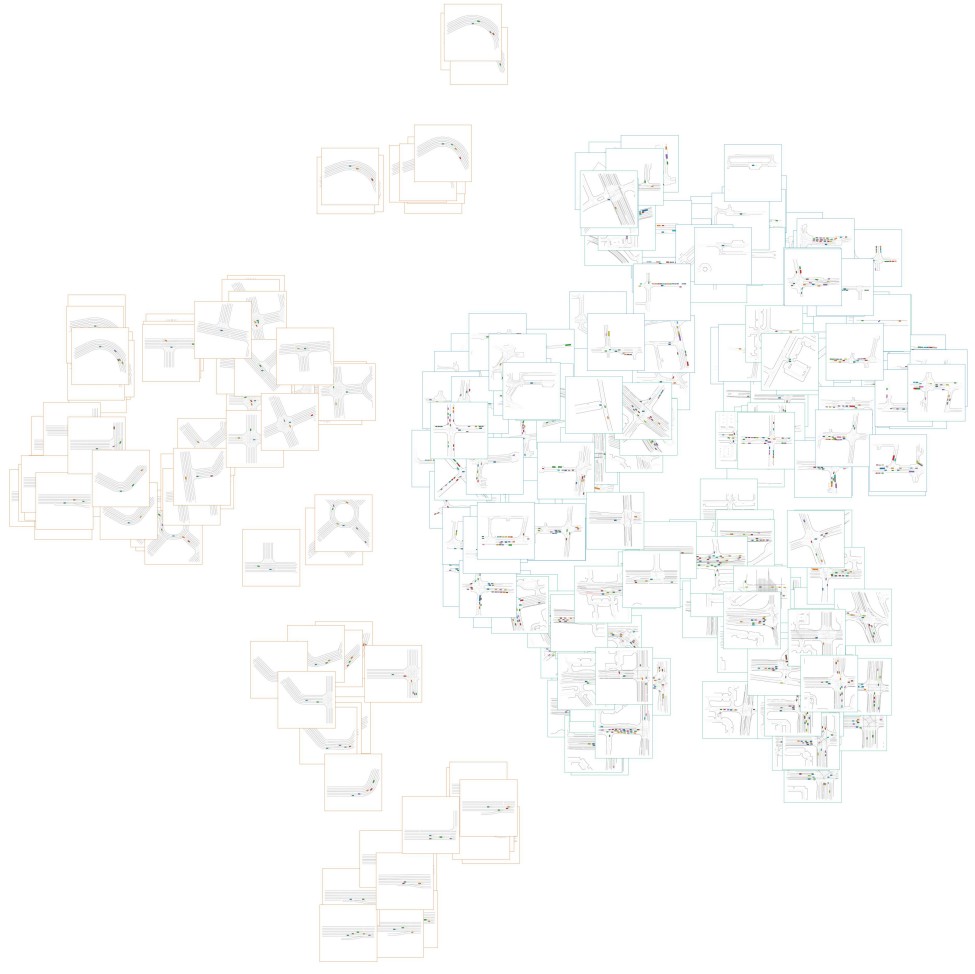

Figure 9: t-SNE visualizations of 3000 scenarios. PG scenarios (□) are located mostly on the left side, while the scenarios from nuPlan (□) and the Waymo scenarios (□) are scattered on the right side.

## A.4  t-SNE Visualizations

The trained model's encoder is used to extract feature embeddings of a given scenario sample. The embeddings are then visualized with t-SNE method to show the similarities and differences. The detailed hyperparameters of t-SNE are shown in Table 5.

The t-SNE result is plotted in Fig. 9. The clustering results show that there is a large domain gap between real-world scenarios (Waymo and nuPlan) and synthetic scenarios (PG). Besides, domain gap exists even in real-world datasets. One prominent feature of the Waymo dataset is the complex crossroad (shown in the lower right corner), which the nuPlan dataset is lacking. The t-SNE visualization reveals the differences between different traffic datasets.

| Table 4: TrafficGen | |
| --- | --- |
| Hyper-parameter | Value |
| Batch Size | 256 |
| Feature Size | 1024 |
| Training epochs | 100 |
| Learning Rate | $3e{-}4$ |
| Activation Function | "relu" |
| MCG Layers | 5 |

| Table 5: t-SNE | |
| --- | --- |
| Hyper-parameter | Value |
| Components Number | 2 |
| Init | pca |
| Learning Rate | auto |
| Perplexity | 30 |
| Early Exaggeration | 12 |

## B  Single-agent Cross-dataset Generalization Experiment

We use the PG database and nuPlan database for cross-dataset generalization experiments. The goal is to investigate how the sim-to-real gap affects the generalizability of the learning-based vehicle controller. To this end, we train agents on synthetic PG scenarios [30] and nuPlan [6] scenarios respectively, and test them on the same held-out real-world test set.

### B.1  Task Setup

To be specific, the task is to follow the trajectory of the data collection car and drive as fast as possible while avoiding collisions.

**Observation.** The observation of the RL agents is as follows:

1. A 120-dimensional vector denoting the Lidar-like point clouds with $50m$ maximum detecting distance centering at the target vehicle. Each entry is in $[0, 1]$ with Gaussian noise and represents the relative distance of the nearest obstacle in the specified direction.
2. A vector containing the data that summarizes the target vehicle's state such as the steering, heading, velocity, and relative distance to the trajectory to follow.
3. The navigation information that guides the target vehicle toward the destination. Concretely, it consists of 10 points sampled on the future trajectory and the distance between two consecutive points is $2m$. The points will be projected to the vehicle coordinates.
4. A 12-dimensional vector denoting the Lidar-like point clouds with $50m$ maximum detecting the boundary of the drivable area, like the solid lines or sidewalks. (Optional)

As the vehicle for collecting nuPlan data in Boston sometimes drives out of the drivable area for bypassing the cones or barriers, crossing the drivable area boundary usually happens. Therefore, we didn't use the boundary detector in our experiments, and crossing the boundaries like solid lines won't terminate the episode nor penalize the agent.

**Action.** The driving policy is a fully end-to-end model and directly controls the low-level throttle and steering angle. The action $a$ is a continuous two-dimensional vector with entries in $[-1, 1]$. By multiplying coefficients and clipping the extreme value, the action will be converted into the engine force and steering angle for changing the vehicle states.

**Reward and Cost Scheme.** The reward function is composed of four parts as follows:

$$R = c_1 R_{disp} + c_2 P_{smooth} + c_2 P_{collision} + R_{term}. \tag{1}$$

The *displacement reward* $R_{disp} = d_t - d_{t-1}$, wherein the $d_t$ and $d_{t-1}$ denotes the longitudinal movement of the target vehicle in Frenet coordinates of the target trajectory between two consecutive time steps, providing a dense reward to encourage the agent to move forward. The *smooth penalty* $P_{smooth} = min(0, 1/v_t - |a[0]|)$ incentives the agent to drive smoothly and avoid a large steering value change between two timesteps, especially, when the velocity is high. $v_t$ and $a[0]$ denote the current velocity and the steering value respectively. In addition, if a collision with a vehicle, human, or object happens at timestep $t$, the agent will receive a *collision penalty*. The penalty is set to $P_{collision} = 2$ when colliding with a human or vehicle and $P_{collision} = 0.5$ for colliding with an object like cones and barriers. We also define a sparse *terminal reward* $R_{term}$, which is non-zero only at the last time step. At that step, we set $R_{disp} = R_{speed} = 0$ and assign $R_{term}$ according to the terminal state. $R_{term}$ is set to $+10$ if the vehicle reaches the destination, $-5$ for being $2.5m$ away from the reference trajectory. We set $c_1 = 2$, $c_2 = 1$ and $c_3 = 1$ .

**Termination Conditions and Evaluation Metrics.** The episode will be terminated only when: 1) the agent drives $2.5m$ away from the reference trajectory 2) the agent arrives at the destination and 3) the agent can not finish the episode in $recorded\_episode\_length + 50$ steps. For each trained agent, we evaluate it in the held-out test environments and define the ratio of episodes where the agent arrives at the destination as the *success rate*. The definition is the same for *out of road* and *Timeout*. For evaluating the driving behavior, the speed in each scenario is collected and then averaged across all scenarios. Also, a metric similar to the *success rate* is measured and called *route completion*, which is the ratio of moving distance to the length of the whole reference trajectory. Since each agent are trained across 5 random seeds, this evaluation process will be executed for 5 agent which has the same training setting but different random seeds. We report the average and std on the metrics mentioned above.

## B.2 Curriculum Training System

In the single-agent experiments, 40,000 scenarios are used for training agents in both PG scenarios and nuPlan scenarios. Loading each scenario from scratch costs a significant amount of time, and thus we would like to buffer the scenarios in RAM for repeated use. However, the memory consumption is nonnegligible, especially when we train 5 policies concurrently and launch 20 workers to collect rollout for each policy. Assuming each scenario consumes about 10MB of memory, buffering 40,000 training scenarios in each worker process consumes 40,000×100×10MB=40,000GB=40TB of memory, which requests a powerful and expensive cluster to train agents in large-scale scenarios. We adopt **curriculum training** scheme for overcoming this issue, which reduces the memory by $99.9\%$.

**Curriculum Training.** We first sort the training scenarios according to the difficulty score calculated by: $track\_length \times cumulative\_curvature$. $track\_length$ is the moving distance of the data collection car. A greater moving distance corresponds to a higher velocity, which in turn indicates a higher difficulty score. This value then will be multiplied by a weight $cumulative\_curvature$ which quantifies the level of bending in the trajectory. After this, scenarios can be divided into 100 levels with 400 scenarios in each level. Only when the *Success Rate* reaches $75\%$, the worker will move to the next level and release the memory used to store scenarios of the previous level. To further reduce memory usage, we split the scenarios in each level into 20 subsets, and each worker only loads scenarios from the corresponding subset. Finally, only $0.05\%$ scenarios (20 scenarios) are actually loaded in each worker, which saves memory by a large margin.

This curriculum training scheme not only makes it possible to finish training on a single server but boosts training efficiency and performance. We conduct an ablation study without the curriculum training, which costs a long time to train. This training scheme releases every scenario after using and reloading it when needed. The inferior performance highlights the necessity of having curriculum training.

## B.3 Results

As shown in Fig. 10, we present experiment results for four training settings: *nuPlan with curriculum training*, *nuPlan without curriculum training*, *PG with curriculum training*, and *Combine PG and nuPlan w/ curriculum training*. The four used databases are *PG-train*, *nuPlan-train*, *PG-nuPlan-train*, and *nuPlan-test*, comprising 40,000, 40,000, 40,000, and 5,000 scenarios respectively. Considering *nuPlan with curriculum training* as the baseline, our analysis focuses on two factors: the data source and the inclusion of curriculum training. The evaluation of real-world scenarios underscores the significance of in-distribution real-world training data and the curriculum training system.

Actually, the poor performance of *nuPlan without curriculum training* is also exposed at the training stage. In spite of increasing the *data coverage*, the agents under this setting always have a poor *training success rate* and *training route completion* throughout the whole training process. However, agents trained in *nuPlan with curriculum training* setting are more stable and have an increasing *training success rate* as they can exploit the scenarios that properly match their current driving ability.

For investigating the influence of data sources, the conclusion can be drawn only from the test results which show that agents trained under *PG with curriculum training* can not generalize to real-world settings well. And the failure mode is that agents trained in synthetic scenarios can not learn to drive at high speed. And they don't move until other vehicles move far from them. This behavior is reflected on the *test speed* as well. However, it doesn't mean that the synthetic data is useless. When

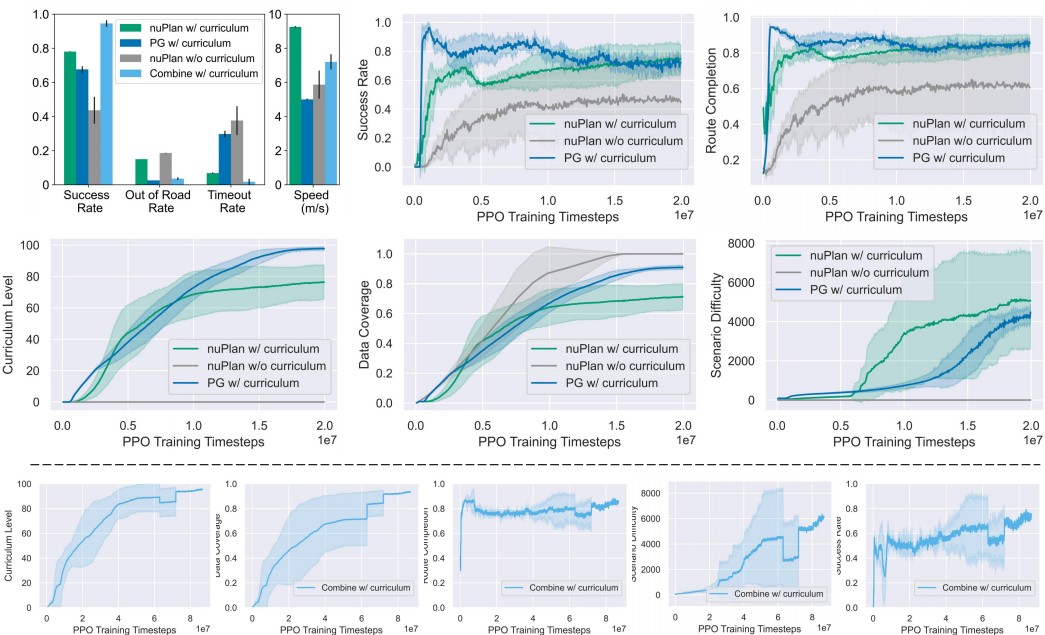

Figure 10: The upper left figure shows the evaluation results for agents trained in 4 different settings. For the remaining training details, we plot the results of three settings together and the other dataset individually. This is because training agents on *Combine w/ curriculum* takes more steps and rollouts to converge and reach the maximum level, compared to training agents on another 3 datasets (2e7 steps vs 8e7 steps). The *curriculum level* is calculated as the average worker level across 5 seeds, with 20 workers per seed. *Data coverage* refers to the proportion of scenarios in the training set that the agent encounters throughout the entire training process. A value of 1.0 indicates that the agent has visited all scenarios. *Scenario difficulty* is determined by calculating the average difficulty score across all scenarios collected during a PPO optimization epoch.

agents are trained on the combined datasets, despite consuming more data, they can actually manifest the best performance. This is because the PG scenarios contain a number of curved roads and thus agents trained in these scenarios can have a better ability to follow a curvature trajectory.

Another interesting phenomenon is the drop of *training success rate* when the *scenario difficulty* and *curriculum level* increase, which is also reported in [9] as well. And this value finally coverage to the test *success rate*. Therefore, we infer that given enough training scenarios, the test performance of learning-based agents can be roughly reflected by the training-time performance.

The hyper-parameters of the RL training are listed in Table 11.

## C   Multi-agent Policy Learning

### C.1   Experiment Setting

We can load the real-world dataset into MetaDrive simulator and create multi-agent interactive policy environment. We first instantiate agents in the environment where their initial states are loaded from the real-world dataset. The initial states include position, heading, velocity and the size. Then, we assign `EnvInputPolicy` to all agents and allow them to be controlled by external RL policies.

The ground-truth (GT) trajectories are not accessible to the learning agents but serve as the supervision via reward function (in RL) or observations (in IL).

Concretely, we create the reward function similar to Eq. 1 with slightly different weights:

$$R = R_{disp} + P_{collision} + R_{term}. \tag{2}$$

The *displacement reward* $R_{disp} = d_t - d_{t-1}$, wherein the $d_t$ and $d_{t-1}$ denotes the longitudinal movement of the target vehicle in Frenet coordinates of the **target trajectory** between two consecutive time steps, provides a dense reward to encourage the agent to move forward. In addition, if a collision with a vehicle, human, or object happens at timestep $t$, the agent will receive a *collision penalty*.

The penalty is set to $P_{collision} = 1$ when colliding with a human or vehicle. We also define a sparse *terminal reward* $R_{term}$, which is non-zero only at the last time step. At that step, we set $R_{disp} = R_{speed} = 0$ and assign $R_{term}$ according to the terminal state. $R_{term}$ is set to $+10$ if the vehicle reaches the destination, $-1$ for being $10m$ away from the reference trajectory. We transform the GT trajectory into the simulator to create the target trajectory and use the displacement reward as the primary supervision from the dataset.

On the other hand, in multi-agent imitation learning we use the GT trajectory to form a dataset of observations and follow the setting of learning from observations or say action-free imitation learning. Specifically, for each frame in the scenario, we load the states of all actors at this frame into the simulator and utilize the sensor simulation functionality of MetaDrive to simulate the observations of each actor. The observation follows Sec. B.1. We form the dataset of observation sequences of all actors in the dataset.

The ground-truth trajectory can be used to measure the learned behaviors. The the route completion rate is the ratio between the length of projected agent trajectory and the length of GT trajectory. The average distance between agent trajectory and the GT trajectory is computed as follows:

$$\frac{1}{T} \sum_{t=1}^{T} ||pos_{agent,t} - pos_{GT,t}||. \tag{3}$$

where $T$ is the minimum length of agent trajectory and the length of GT trajectory. Therefore, for the agents that terminate quickly after spawning, the average distance will be quite small. The final distance is computed as distance between the last position of GT trajectory and the last position of agent trajectory. The cost is the number of crashes of an agent in one episode. There are two terminal conditions. If the route completion rate exceeds 95%, the agent is marked successful. If the agent moves out of $10m$ aways from the reference trajectory, the agent is marked out of road and failed. We don't terminate agent's episode if it crash with other objects.

## C.2 Baseline Details

**MA-GAIL**. We use GAIL [25] in multi-agent setting [46] but the discriminator distinguishes state-next state pair, instead of state-action pair [50]. We use PPO as the underlying RL algorithm in GAIL. The hyper-parameters are listed in Table 6.

**MA-AIRL**. We also train multi-agent Adversarial Inverse RL [65] (AIRL) with an additional inverse dynamics model for estimating the expert actions. The inverse dynamics model is trained concurrently with the AIRL policies and learns the action given state-next state pairs from the environment interactions. AIRL will learn a reward function and use the reward function to build a discriminator as GAIL. We also use PPO as the underlying RL algorithm. The hyper-parameters are listed in Table 7.

**MARL baselines**. We train independent PPO [43, 41] and TD3 [23] agents as well as the Coordinated Policy Optimization (CoPO) agents [39] as MARL baselines. The hyper parameters are given in the next section.

# D  Hyper-parameters

Table 6: Action-free Multi-agent GAIL

| Hyper-parameter | Value |
| --- | --- |
| Discriminator LR | 5e-5 |
| Discriminator L2 Norm | 1e-5 |
| Discriminator SGD Num Iters | 1 |
| Discriminator SGD Minibatch Size | 1024 |
| PPO SGD Minibatch Size | 512 |
| PPO Batch Size | 2000 |
| PPO LR | 1e-4 |
| PPO SGD Num Iters | 10 |
| PPO Clip Parameter | 0.2 |
| PPO Lambda | 0.95 |
| PPO Gamma | 0.99 |

Table 7: Action-free Multi-agent AIRL

| Hyper-parameter | Value |
| --- | --- |
| Discriminator Loss LR | 3e-4 |
| Discriminator L2 Norm | 1e-5 |
| Discriminator SGD Num Iters | 5 |
| Discriminator SGD Minibatch Size | 1024 |
| Inverse Dynamics SGD Num Iters | 100 |
| Inverse Dynamics SGD Minibatch Size | 512 |
| Inverse Dynamics LR | 1e-4 |
| PPO SGD Minibatch Size | 512 |
| PPO Batch Size | 2000 |
| PPO LR | 1e-4 |
| PPO SGD Num Iters | 10 |
| PPO Clip Parameter | 0.2 |
| PPO Lambda | 0.95 |
| PPO Gamma | 0.99 |

Table 8: Multi-agent PPO

| Hyper-parameter | Value |
| --- | --- |
| PPO SGD Minibatch Size | 512 |
| PPO Batch Size | 2000 |
| PPO LR | 1e-4 |
| PPO SGD Num Iters | 10 |
| PPO Clip Parameter | 0.2 |
| PPO Lambda | 0.95 |
| PPO Gamma | 0.99 |

Table 9: CoPO

| Hyper-parameter | Value |
| --- | --- |
| LCF LR | 1e-4 |
| LCF Num Iters | 5 |
| Neighborhood Distance | 40 m |
| PPO SGD Minibatch Size | 512 |
| PPO Batch Size | 2000 |
| PPO LR | 1e-4 |
| PPO SGD Num Iters | 10 |
| PPO Clip Parameter | 0.2 |
| PPO Lambda | 0.95 |
| PPO Gamma | 0.99 |

Table 10: TD3

| Hyper-parameter | Value |
| --- | --- |
| Critic LR | 1e-4 |
| Actor LR | 1e-4 |
| Tau for Target Update | 5e-3 |
| Train Batch Size | 100 |

Table 11: Single-Agent PPO

| Hyper-parameter | Value |
| --- | --- |
| KL Coefficient | 0.2 |
| $\lambda$ for GAE [42] | 0.95 |
| Discounted Factor $\gamma$ | 0.99 |
| Number of SGD epochs | 20 |
| Train Batch Size | 50,000 |
| SGD mini batch size | 200 |
| Learning Rate | 1e$-$4 |
| Clip Parameter $\epsilon$ | 0.2 |
| Activation Function | "tanh" |
| MLP Hidden Units | [512, 256, 128] |
| MLP Layers | 3 |

# E   AD stack testing

## E.1   ROS bridge

As shown in Fig. 11, We provide a ROS bridge along with the *ScenarioNet*, allowing users from the ROS community to develop and test their systems with massive real-world data. As shown in Fig. 12, information like camera, lidar, and mid-level representations can be retrieved from the simulation.

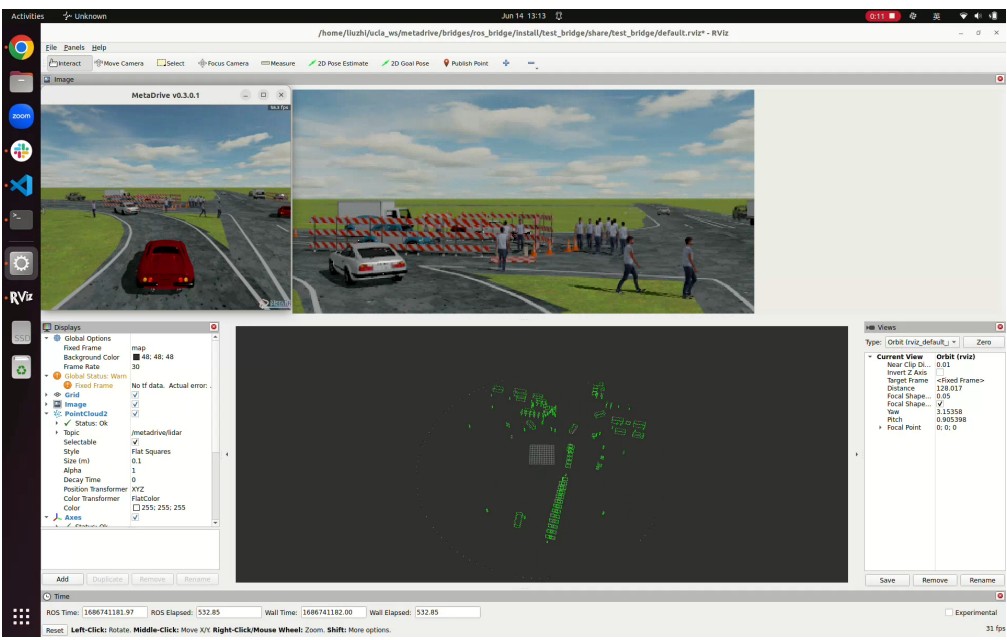

Figure 11: The interface of ROS bridge allowing connecting *ScenarioNet* and ROS community.

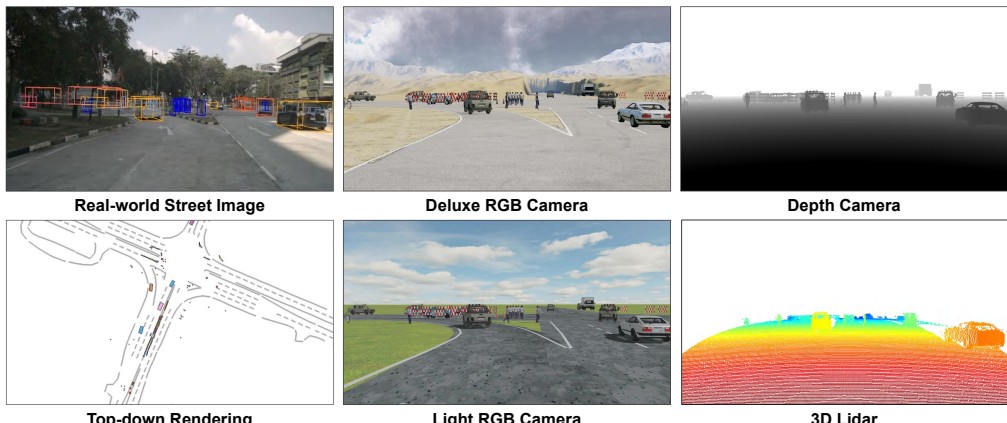

Figure 12: Multimodal sensory data provided by *ScenarioNet*.

*ScenarioNet* provides not only mid-level scenario representations but multiple sensor outputs like top-down view, RGB camera, depth camera, and cloud points. The deluxe RGB camera is supported by the deferred rendering pipeline. This figure shows the `scene-0061` from nuPlan-mini-split and its digital twin counterparts. It is worth investigating how to reconstruct meshes from the recorded lidar cloud points and images so that we can simulate the sensor output from new views different from the recorded ones. Besides, it is a promising topic to study the **closed-loop** sensor fusing and learning-based control together, which can be supported by *ScenarioNet*.

## E.2 Qualitative Results of Openpilot test

Openpilot [10] is an end-to-end solution for driver assistance. Therefore, a platform providing 3D rendering is necessary, if one would like to study or test the system. *ScenarioNet* is the only one that provides both real-world scenarios and 3D graphics support. As its navigation module is still a beta version, we only test Openpilot in scenarios without diverged roads. We build a small database containing mainly lane-keeping scenarios for conceptually demonstrating that *ScenarioNet* can connect with commercial AD stack. As shown in Fig. 13, the Openpilot system is robust to common scenarios like turning right, lane-keeping, stopping at traffic lights, and passing traffic lights. The demo video is available at https://youtu.be/KjlPBOnCTvg

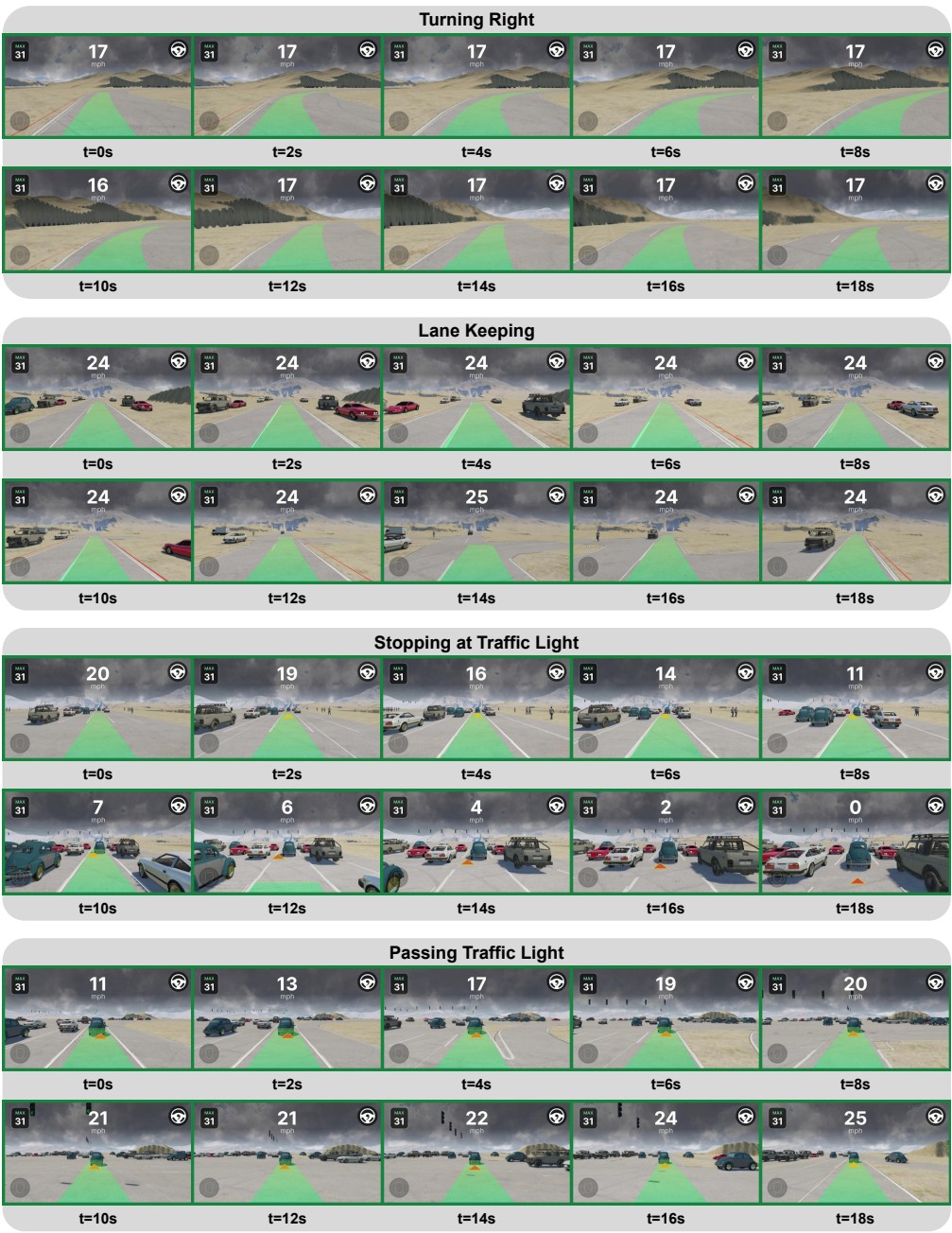

Figure 13: Openpilot manages to overcome four representative scenarios.

# F    Scenario Databases

In the database statistics Table, the *Intersection Ratio* is the ratio of scenarios having traffic lights for real-world data, and the ratio of scenarios containing intersections and roundabouts for synthetic data.

## F.1    Waymo Database

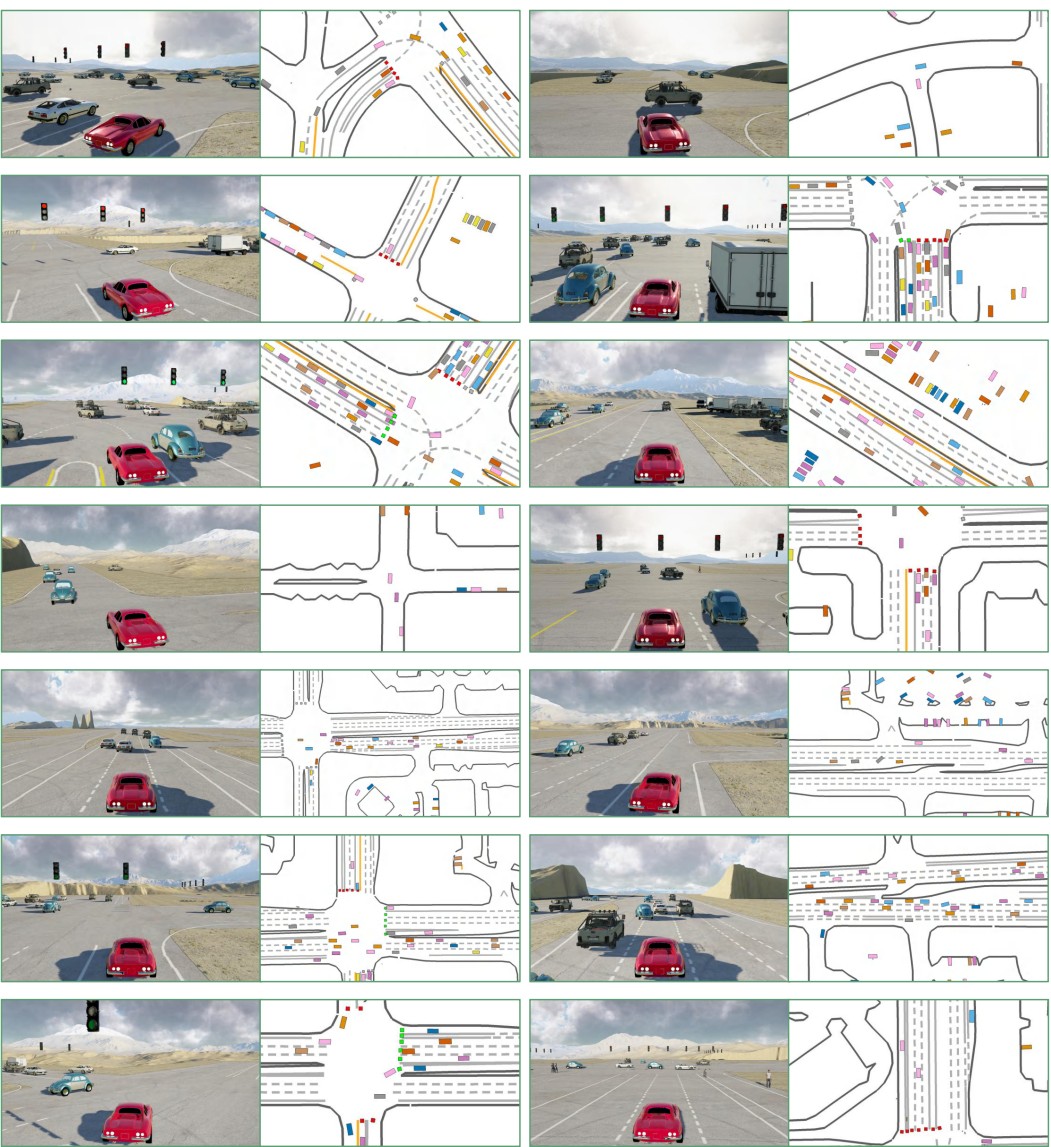

Figure 14: Rendered traffic scenarios and their top-down views from the Waymo database.

We construct the Waymo database from the **20 seconds** version [47] motion data with Google Cloud path at `waymo_open_dataset_motion_v_1_2_0/uncompressed/scenario/training_20s`. As Waymo data is collected with the altitude calibrated, we can filter the overpass scenarios by excluding the scenarios with significant changes in height. In addition, we exclude the scenarios where the ego car waits at the red light for a long time by selecting scenarios where the ego car moving distances are greater than 10 meters. We provide the rendered scenario examples from this dataset in Fig. 14. Top-down views show that Waymo scenarios contain diverse road structures and a large number of vehicles.

## F.2 nuPlan Database

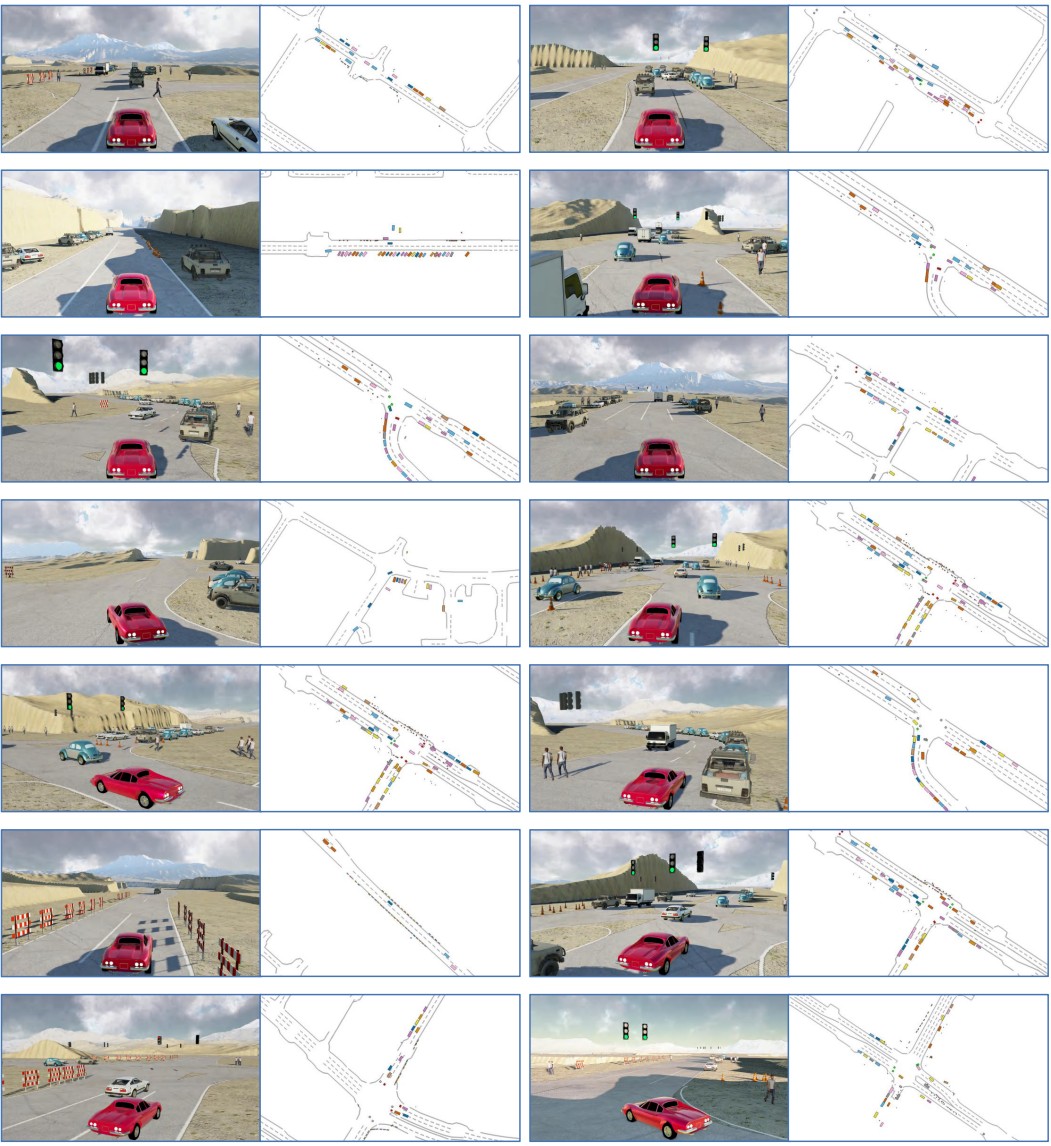

Figure 15: Rendered traffic scenarios and their top-down views from the nuPlan database.

According to `https://www.nuscenes.org/nuplan`, nuPlan has more than 1500 hours of driving data collected in 4 different cities: Las Vegas, Singapore, Pittsburgh, and Boston, here we only use the data collected in Boston as we want to keep all databases in experiments to have similar sizes. Data collected in other cities can also be converted to our scenario format.

We use the V.1.0 version data, which contains approximately 50,000 scenarios after excluding scenarios where the ego car moving distances are less than 10 meters. It is noticeable that nuPlan updated the data to version v1.1 recently (one week before the NeurIPS 2023 dataset track deadline), which may induce some differences if building a database from this new version. As shown in Fig. 15, the scenario examples reveal that *nuPlan Boston split* has cluttered scenes and contains many traffic cones and barriers besides vehicles. We additionally find that in some nuPlan scenarios, the ego car trajectory is out of the drivable area for sidestepping the barriers or cones.

## F.3 PG Database

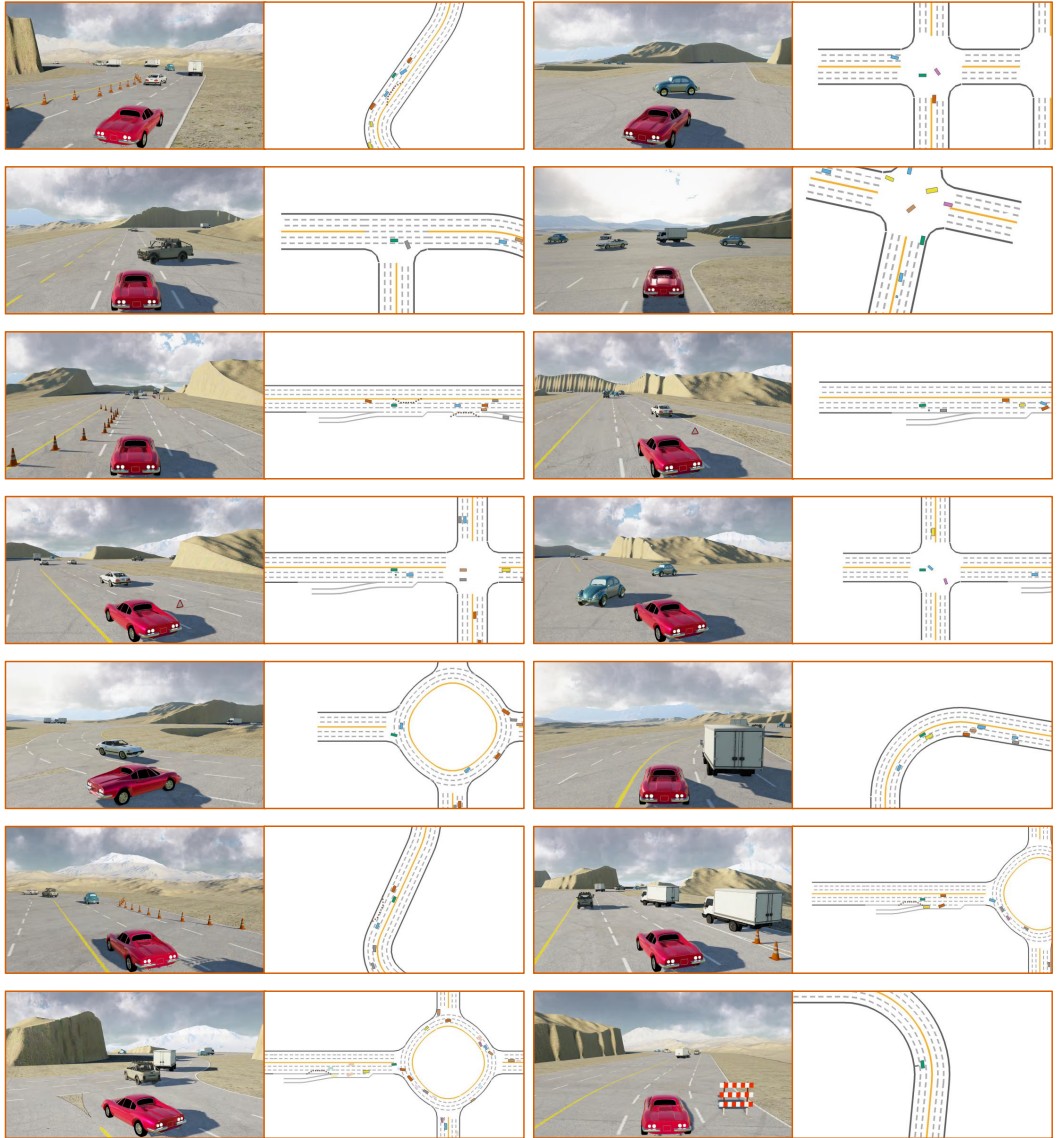

Figure 16: Rendered traffic scenarios and their top-down views from the PG database.

Unlike the previous two datasets collected in the real world, PG scenarios are synthesized according to a set of rules. For map generation, two blocks are sampled from a set of candidate roadblocks and connected to form a map. Those blocks include intersections, roundabouts, straight roads, curved roads, Ramp, and so on. Once the map is defined, a traffic generation rule will be applied to scatter vehicles and road objects like traffic cones on the map. The detailed scenario generation config such as the block distribution for sampling can be found at https://github.com/metadriverse/metadrive/blob/0a929f8130b34e4428067390f20f872d1d6d224a/metadrive/component/algorithm/blocks_prob_dist.py#L4. All vehicles choose a destination automatically and be actuated by IDM policy which can keep a proper distance from the front vehicle and perform a lane change when the front object is static. The scenario data is collected by an IDM policy as well. Some scenario examples are shown in Fig. 16.

# G  Discussion: how to improve visual fidelity?

The visual fidelity of the closed-loop simulation can be further improved with the collected raw sensor data. We already have some plans to improve it and will include this discussion in the paper for sharing our thoughts regarding realistic sensor simulation.

Currently, the sensor simulation including the camera and lidar is achieved in a Computer Graphics (CG) way, where people try restoring the mesh and texture for objects from real-world data and shading these models based on lights and materials to make them visually realistic. To improve the rendering results of *ScenarioNet*, we do plan to reconstruct the geometry data for traffic participants and objects from the driving videos. Besides, we also consider connecting *ScenarioNet* with Unreal Engine or Nvidia Omniverse in the future as they could provide better shading results.

On the other hand, NeRF is an alternative to improve the quality of sensor simulation. Through volume rendering, it can directly synthesize new camera views and point clouds from the driving videos when traffic participants and the ego car move with different trajectories and poses in the closed-loop training. This way is purely data-driven and can exempt the need for restoring 3D assets like objects and buildings. Recent results [61, 64, 60] already demonstrate its potential in terms of camera and lidar simulation. However, how to make the NeRF scene editable is still an open problem; hence, we plan to investigate this in the future.

