# OpenReview forum: "ScenarioNet: Open-Source Platform for Large-Scale Traffic Scenario Simulation and Modeling"
_NeurIPS.cc/2023/Track/Datasets_and_Benchmarks — NeurIPS 2023 Datasets and Benchmarks Poster_

### Official Review · Reviewer_Ei6A · 2023-07-21
**Review of ScenarioNet: Open-Source Platform for Large-Scale Traffic Scenario Simulation and Modeling.**

**Rating:** 7
**Confidence:** 3

**Strengths:**

### Main strengths

#### Framework
The implementation of ScenarioNet which is based on a unified scene description that serves as a bridge to utilize multiple datasets under a common framework. The framework is structured in a modular way, allowing users to start with just the data construction module and progress to running a 3D visualization of single or multi-agent training policies. This flexibility empowers researchers to adapt the platform to their specific needs.

### Additional contributions in the paper:

#### Benchmarks for Autonomous Driving Evaluation.
The ScenarioNet platform serves as a benchmark for evaluating the safety of autonomous driving systems in simulation before their real-world deployment. By using data-driven scenarios and diverse traffic behaviors from real-world datasets, researchers can assess the performance and robustness of autonomous driving stacks more effectively in a simulated environment.

#### Advancements in Machine Learning and Autonomous Driving.
The authors demonstrate how ScenarioNet facilitates several research opportunities in machine learning and autonomous driving. It supports large-scale scenario generation, imitation learning, and reinforcement learning tasks in both single-agent and multi-agent settings.


**Additional Feedback:**

N/A (all major comments were addressed in the previous sections)

**Clarity:**

### Paper clarity.
The paper is generally well written with some exceptions about explanations related to the flexibility of the system, technical limitations and Figures 4 and 5, as already pointed out before.


**Correctness:**

### Paper claims
In general the claims are adequate in relation to the framework and the novel unified scene description proposed. As already mentioned, perhaps a more detailed discussion on its flexibility would be interesting, as authors want to propose this as some sort of new standard for scene representations. Limitations should be clearly stated as otherwise it may be frustrating for other researchers adopting the format finding them later by themselves.


**Documentation:**

It would be beneficial to have documentation readily available to ensure a smoother understanding and utilization of the system. For example providing some examples of use cases with a step-by-step explanation.


**Ethics:**

Up to my knowledge on this topic, I haven’t identified any.


**Limitations:**

There was not a clear discussion about the limitations, just a small paragraph about the potential lack of variety in the current 3D assets and a disclaimer about safety regulation. It seems short given the software engineering nature of this project, that no other possible bottlenecks in processing datasets were identified (i.e. parallelism, multi-GPU support, etc.) and also potential (but understandable) limitations related to the scope of this project (simulation, interactive visualization, etc.).


**Opportunities For Improvement:**

#### Clarify the Role of ScenarioNet and its flexibility.
Elaborate on how the proposed unified scene description format helps in integrating and leveraging data from various driving datasets. It should be clarified how other users can extend this initial scheme to support other potentially required features. The claim of flexibility requires a clear explanation about how researchers can customize the platform and what benefits it offers in different research scenarios. Also, could authors better clarify the relationship among ScenarioNet and TrafficSim? Could TrafficSim be used for free as ScenarioNet? It seems widely mentioned throughout the paper but nothing is said about the licensing or availability of TrafficSim, e.g. in Section 5.

#### Quantify Advancements and better benchmarking.
When discussing the advancements in machine learning and autonomous driving facilitated by ScenarioNet, consider providing specific metrics or results to demonstrate the platform's effectiveness. Mention any comparative studies or performance improvements achieved using the platform. The only quantitative results are presented in Table 3 and measurement units were omitted (thus, not well explained). The authors may want to better explain the evaluation criteria, their testing methodologies (specially for AD stacking), and how the platform addresses the challenges of benchmarking in autonomous driving research.

#### Visual analysis (Sec. 4.2).
While t-SNE could be particularly useful for visualizing complex datasets and revealing underlying structures and patterns. However, t-SNE is a non-deterministic algorithm, meaning that the results can vary depending on the random initialization and various hyperparameters chosen during the process. Two different runs of t-SNE with different parameters can lead to significantly different visualizations, even though they may originate from the same dataset.

#### Figure 5.
It is weird that authors mix success rate and timeouts in the same figure, as they probably need to use different measurement units. Also, could authors explain why they decided to leave “PG w/o curriculum” out of those comparisons?

#### Minor typo.
“Nivdia” instead of “Nvidia” (l38 of the appendix)

#### Potential Challenges and Limitations.
As a suggestion, the authors just mention potential lack of variety in the 3D assets and safety disclaimers, but no technical limitations are described at all. It seems strange that no other technical limitations are highlighted here, at the very least in terms of performance and processing bottlenecks, or any constraints imposed by the simulation environment. If applicable, it would be interesting to highlight if there are any user-friendly interfaces or tools within ScenarioNet that make it easier for researchers to work with the platform. This could include intuitive interfaces, comprehensive documentation, or sample code examples.


**Relation To Prior Work:**

### Related work.
This reviewer believes the discussion provided fairly covered the previous work alternatives.


**Summary And Contributions:**

### Summary
This submission presents an open-source platform for scenario integration and simulation of large-scale driving datasets. The proposed framework proposes a unified scenario description that allows multiple operations to ease the integration of different data sources, i.e. data coming from procedurally-simulated scenarios or data logs coming from real-world driving databases. The framework allows raw data conversion, sanity-checks, merging, splitting, sampling and filtering operations, thus producing new curated databases represented in their proposed “unified scene description”. In addition, their framework can be used to feed all this new data into an upgraded version of the MetaDrive simulator, which supports physics simulation the Bullet physics engine. Authors run a series of experiments (detailed in the Appendix) which demonstrates several applications of the framework: 4.1) a new database construction; 4.2) a visualization using t-SNE of multiples scenarios in a common embedded space; 4.3) an experiment on RL policies and 4.4) a multi-agent cooperative learning experiment in the MetaDrive simulator; and v) an AD stack test based on a ROS bridge implementation connecting multiple software stacks with en end-to-end AD system (OpenPilot).

### Contributions
The main contributions of this work are:
####  A new framework
The purpose of the newly proposed framework, ScenarioNet, is to integrate large-scale driving datasets. Authors provide an open framework and propose a unified scene description that contributes to facilitate better experiments by the research community in autonomous driving for perception tasks like 3D detection and/or trajectory forecasting.
#### Closing the gap to use real-data as source data for simulation.
Their main idea is to collect, aggregate and filter a massive number of complex traffic scenarios (i.e. based on real-data driving logs, procedural simulations, etc.) and facilitate the creation of digital twin counterparts through simulation. The use of data-driven scenarios potentially allow for a more accurate and diverse simulation environment, facilitating machine learning research and autonomous driving development. ScenarioNet provides researchers the possibility of using real-world scenarios that can be replayed and interacted for varied purposes.

---

> ### Author Response · Authors · 2023-08-09
> **Response to Reviewer Ei6A**
>
> Thank you very much for your time and the detailed review! We prepare detailed responses to your questions. In particular, we include detailed discussions on the limitation. Please find the detailed response below.
>
> ---
>
> > **Q1: Clarify the Role of ScenarioNet and its flexibility. It should be clarified how other users can extend this initial scheme to support other potentially required features.**
>
> **A1:** The internal scenario description is actually a `dict` object with some mandatory fields. When converting original driving data to the internal format of *ScenarioNet*, our batch converting tools will automatically check if these fields are filled to guarantee that these scenarios are runnable. With all mandatory fields filled, users can add any number of new fields to the `dict` describing the scenario. For example, real-world driving data usually contains only state trajectories without expert actions, while researchers can not study Imitation Learning without action labels. In this way, they can extract the possible expert action with inverse dynamics and store them in the scenario description as well.
>
> ---
>
> > **Q2: Could authors better clarify the relationship between ScenarioNet and TrafficSim? Could TrafficSim be used for free as ScenarioNet?**
>
> **A2:** TrafficSim is a data-driven method that simulates traffic given an initial traffic state. However, TrafficSim is trained and evaluated on ATG4D which is an in-house dataset. Thus *ScenarioNet* provides infrastructures for researchers who plan to replicate TrafficSim. On the other hand, the well-trained TrafficSim model can make the traffic vehicles reactive to achieve closed-loop simulation. Compared to directly replaying the logged trajectory, controlling traffic vehicles with TrafficSim allows rich interactions with the ego car like yielding, merging, and braking behaviors. As stated in *Section 3.3, Control Policies*, *ScenarioNet* provides the `policy` interface to determine how to control each object in the scene. Accordingly, it is easy to connect TrafficSim and the simulation module of *ScenarioNet* for making the traffic vehicles realistic and reactive.
>
> ---
>
> > **Q3: Please provide specific metrics or results to demonstrate the platform's effectiveness. Mention any comparative studies and explain the evaluation criteria, their testing methodologies (especially for AD stacking)**
>
>
> **A3:** We demonstrate the effectiveness of *ScenarioNet* and how it can facilitate AD research with several experiments including single-agent/multi-agent RL, traffic generation, and AD stack testing. For example, in the single RL experiment, the agents trained on nuPlan scenarios outperform those trained on synthetic scenarios when benchmarking on held-out nuPlan test scenarios. Moreover, this sim-to-real gap is revealed by the traffic generation experiments as well. These results suggest that the real-world driving scenario reconstruction enabled by *ScenarioNet* is important to realistic driving policy learning and traffic generation. In addition, the single-agent/multi-agent RL experiment settings are well introduced in the Appendix including the measure units, learning curves, and training schemes. For AD stack evaluation, we provide qualitative results in Appendix and a demo video on our webpage: https://youtu.be/KjlPB0nCTvg
>
>
> ---
>
> > **Q4: Two different runs of t-SNE with different parameters are required**
>
> **A4:** We provide two more runs of t-SNE results at https://drive.google.com/file/d/1oUbcWXVi_fPnPVqAG8rJlOx3fv0zYvkt/view?usp=sharing. Both of them and the one in the paper reach the same conclusion that the sim-to-real gap exists.
>
> ---
>
> > **Q5: In Figure 5, probably success rate and timeout rate need to use different measurement units. Also, could the authors explain why they decided to leave “PG w/o curriculum” out of those comparisons?**
>
> **A5:** Both Success Rate and Timeout Rate are the proportions of the number of scenarios out of all tested scenarios where the agent reaches the destination or cannot reach within a pre-defined horizon. Thus they are all percentages and share the same units. We didn’t include the result of *PG w/o curriculum* as it is not informative. This experiment aims to show that 1) curriculum learning and 2) data sources are important for training a driving policy that can perform well in real-world scenarios. Therefore, we vary these two factors respectively to illustrate this based on the *nuPlan w/ curriculum* experiment. Varying both factors together can’t provide insights so we don’t include it.
>
> ---
>
> > **Q6: No technical limitations are described at all like the performance and processing bottlenecks, or any constraints imposed by the simulation environment.**
>
> **A6:** Please see *Common Response, Technical Limitations*.
>
> ---
>
> > **Q7: It would be beneficial to have documentation.**
>
> **A7:** We agree with the reviewer. The detailed documentation development plan can be found at *Common Response, Documentation*.

---

> > ### Comment · Reviewer_Ei6A · 2023-08-25
> > **Appreciate authors response and final reminders**
> >
> > I would like to thank the authors for their effort in answering all the questions raised during my original review.
> >
> > I appreciate the extended discussion about potential limitations of the system in terms of parallel processing. For the sake of clearness, it still remains unclear to me whether the current CPU implementation can exploit multi-threading or is still a single-thread simulation.
> >
> > Regarding Fig. 5, I still slightly disagree with the choice of merging velocity with success or out-of-road ratios since they fundamentally represent different concepts. But at least I better understand now the figure and what motivated their choice.
> >
> > Finally, while the authors refer to *https://scenarionet.readthedocs.io/en/latest/* for new documentation appearing during the rebuttal period, For now the website seems just en entry webpage to fill in the future. Just kindly remind them, that they have promised: "to provide details about the internal scenario description, API reference, system design, and guidelines for third-party contribution. Also, illustrative examples will be included such as importing new datasets, running simulations, and training models."
> >
> > Overall, I would maintain my original support for the acceptance of this paper as part of NeurIPS 2023 Track Datasets and Benchmarks.
> >
> > Best regards

---

> > > ### Author Response · Authors · 2023-08-28
> > > **Thank you for the reply**
> > >
> > > Dear reviewer,
> > >
> > > Thank you very much for your time and further discussion. Please find our responses to your remaining concerns below:
> > >
> > > 1. The current simulation implementation is **single-threaded** due to the limitations of the Python GIL. Therefore, we adopted multi-process tools such as ray/rllib[1] for parallel simulation. We will clarify this further in the paper.
> > >
> > > 2. Unlike the *mean velocity*, which is calculated using the entire episode data, both *success rate* and *out of road* serve as termination conditions. We agree that combining them can be confusing. The primary reason for this approach was to create a more compact graph. We will consider dividing this graph to enhance its readability.
> > >
> > > 3. Thank you for your feedback. The readthedocs system experienced some build errors; however, we have now fixed these issues. The documentation page is up-to-date and available at https://scenarionet.readthedocs.io/en/latest/. Here's the progress since our initial response (x for finished, blank for todo):
> > >     - Installation guide [x]
> > >     - Minimal example [x]
> > >     - Operation (API) reference [x]
> > >     - Detailed guide for preparing raw data [x]
> > >     - New dataset support - [x]
> > >     - Simulation [ ]
> > >     - Scenario description [ ]
> > >     - Model training [ ]
> > >
> > >
> > > We greatly appreciate the reviewers' time and thorough review and always welcome suggestions to enhance this submission.
> > >
> > > Best regards,
> > > Authors
> > >
> > > ---
> > >
> > > [1] Moritz, Philipp, et al. "Ray: A distributed framework for emerging {AI} applications." 13th USENIX symposium on operating systems design and implementation (OSDI 18). 2018.

---

### Official Review · Reviewer_KCPD · 2023-07-21
**Unifying Datasets and Training Paradigms for Autonomous Driving**

**Rating:** 6
**Confidence:** 3
**Correctness:** The technical descriptions of the sim…

**Strengths:**

The authors explain their framework in granular detail in the main text and technical appendix.
They have also provided qualitative and quantitative evaluations for some forecasting tasks.
The framework offers a closed loop evaluation with both nonreactive and reactive neighboring agents.

**Additional Feedback:**

N/A

**Clarity:**

There are some minor grammatical errors (e.g., line 133 pg 4, line 231 pg 6).
Section 4.5 states that the Openpilot model passes "all tests without 0 collisions" instead of "with 0 collisions".
Also "procedurally synthetic" on line 180 pg 5.

**Documentation:**

The authors have provided sufficient detail regarding data collection, maintenance, and ethics. In comparison to existing SOTA like nuPlan, the documentation of the code in the repository is a bit lacking.

**Ethics:**

Synthetic 3D models are used to simulate real and procedurally generated scenarios, protecting the privacy of individuals. These scenarios are sourced from publicly available datasets.

**Limitations:**

The value of the procedurally generated data has not been fully demonstrated. Section 4.3 shows how it unable to generalize to nuPlan's test dataset: PG training learns slower movement, has a higher timeout rate, and has a lower success rate. Ideally, it should be demonstrated that joint nuPlan-PG training has a synergistic effect and achieves better scores on at least some metrics than nuPlan alone.

**Opportunities For Improvement:**

Figure 4 illustrates the domain gap between real and simulated data and depicts representative scenarios in clusters across the latent space. I recommend depicting scenarios specifically around the 2 simulated scenes that are clustered with the real scenes. It would be insightful to see which scenarios bridge the sim2real gap.
The authors showcase RL and MARL methods, but not imitation learning methods evaluated using common evaluation metrics such as ADE/FDE.

**Relation To Prior Work:**

The authors compare the features of the proposed framework with several SOTA frameworks, clearly showing how the proposed work differs.

**Summary And Contributions:**

ScenarioNet enables learning for an array of tasks in autonomous driving (scenario generation, imitation learning, reinforcement learning, single- and multi-agent settings) and bridges the gap between different datasets (L5Kit, Waymo, nuPlan, nuScenes).

---

> ### Author Response · Authors · 2023-08-09
> **Response to Reviewer KCPD**
>
> Thank you very much for providing suggestions for improving the clarity and quality of our paper! Apart from the paper revision and typo fixing, we run some new experiments and have the following responses to your questions.
>
> ---
>
> > **Q1: I recommend depicting scenarios specifically around the 2 simulated scenes that are clustered with the real scenes. It would be insightful to see which scenarios bridge the sim2real gap.**
>
> **A1:** Good suggestion! We visualized the two outliers and found that both scenarios happen in the intersection. As real-world scenarios near the 2 synthetic scenarios are also intersection scenarios, we conclude it is the intersection scenarios that bridge the sim2real gap. It is an intriguing finding. We already update this figure in the paper to exhibit these two outliers and add a discussion about it.
>
> ---
>
> > **Q2:  Imitation learning methods evaluated using common evaluation metrics such as ADE/FDE.**
>
> **A2:** For conducting imitation learning, action labels should be extracted from the state transitions with inverse dynamics[1]. But our vehicle model is not as simple as the widely used bicycle model, making it hard to generate expert action labels. Thus we don’t provide an imitation learning baseline. An alternative is to train an imitation policy predicting the next state instead of the action and employ a low-level controller to reach the predicted state[2]. We will try both ways in the future and provide baseline results measured by ADE/FDE.
>
> [1] Vinitsky E, Lichtlé N, Yang X, et al. Nocturne: a scalable driving benchmark for bringing multi-agent learning one step closer to the real world[J]. Advances in Neural Information Processing Systems, 2022, 35: 3962-3974.
>
> [2] Caesar H, Kabzan J, Tan K S, et al. nuplan: A closed-loop ml-based planning benchmark for autonomous vehicles[J]. arXiv preprint arXiv:2106.11810, 2021.
>
> ---
>
> > **Q3: Ideally, it should be demonstrated that joint nuPlan-PG training has a synergistic effect and achieves better scores on at least some metrics than nuPlan alone.**
>
> **A3:** We conduct the suggested experiment. It shows that combing the two datasets for training indeed improves the performance of agents in terms of *Success Rate*, when evaluating them on the same *nuPlan-test* split. A detailed discussion can be found in *Common Response, New Experiment*. We also included the results in our paper by updating the figure and discussing the new experiment.

---

### Official Review · Reviewer_Wnbm · 2023-07-22
**Platform to combine multple autonomous vehicle dataset**

**Rating:** 6
**Confidence:** 3
**Correctness:** The claims seem to be correct.
**Clarity:** The quality of the writing is good.

**Strengths:**

The strength is the combined dataset to a uniform description and show case the benefit of combined dataset on performance evaluation. Future researchers can follow the guideline to provide dataset that's more universal and be used together.

**Additional Feedback:**

it would be great to combine real and simulated sensor data together.

**Documentation:**

The documentation is sufficient for readers to understand and re-produce.

**Ethics:**

No ethics concern.

**Limitations:**

The limitation and societal impact are discussed.

**Opportunities For Improvement:**

Tools to compare the real sensor data vs synthesized one. The platform seems to be generating dataset from the simulator, which is not real dataset, so not sure how this will different from just extract all the scenarios from the AV dataset and apply them in a simulation and retrieve the data. While combined real sensor data can provide more scenarios, a well-engineered simulator can potentially have more scenarios.

**Relation To Prior Work:**

The relation and difference to prior work is clearly discussed.

**Summary And Contributions:**

This paper describes a platform that can combine multiple autonomous dataset and to be used together for a larger variety of the scenarios which can potentially enable higher generalization for complex environments. The contribution lies in the combined dataset of some popular autonomous vehicle dataset and can be used to test the algorithm.

---

> ### Author Response · Authors · 2023-08-09
> **Response to Reviewer Wnbm**
>
> Thank you for your precious time and comments on how we can improve the sensor simulation. Please find the detailed response below.
>
> ---
>
> > **Q1: Tools to compare the real sensor data vs synthesized one.**
>
> **A1:** Our teaser video can reflect the similarity between the synthesized sensor output and the real one: https://metadriverse.github.io/scenarionet/. We also agree with the reviewer and believe that the difference between synthesized sensor data and the real one can be calculated by replaying the reconstructed scenario and running the comparison frame-by-frame with tools like ViSiL[1]. These tools can provide a good metric to indicate the quality and realism of the reconstructed scene in terms of visual appearance and thus would be helpful if we are going to optimize the visual appearance of the simulated scene.
>
> [1] Kordopatis-Zilos, Giorgos, et al. "Visil: Fine-grained spatio-temporal video similarity learning." Proceedings of the IEEE/CVF international conference on computer vision. 2019.
>
> ---
>
> > **Q2: The platform seems to be generating datasets from the simulator, which is not a real dataset, so not sure how this will differ from just extracting all the scenarios from the AV dataset and applying them in a simulation and retrieving the data.**
>
> **A2:** The main reason we render the scene and simulate sensor output rather than using those collected in the real world is that we are doing closed-loop simulation. If we run an open-loop simulation where all objects are moving by following the pre-recorded trajectories, the real sensor data is definitely better than the synthetic one. However, most of the time *ScenarioNet* runs closed-loop simulations where ego cars may have different driving behaviors and affect surrounding traffic vehicles. In this case, the trajectories of the ego car and others may be different from the recorded ones. As a result, original real-world sensor data will be invalid due to the change of scene. In this case, new camera views and cloud points need to be synthesized to accommodate these changes by rendering the different scenes from a new viewpoint.
>
> However, it is still possible to use real-world driving sensor data to improve the sensor simulation quality for closed-loop simulation. We believe it is a promising direction to explore and a detailed discussion about this can be found in *Common Response, How to improve visual fidelity*.

---

### Official Review · Reviewer_DPk1 · 2023-07-25
**Strong concept, interoperability would be even better**

**Rating:** 7
**Confidence:** 3

**Strengths:**

The main contribution lies in the automated loading, converting and handeling of massive real-world scenarios from different sources.
This capability allows to utilize a vastly greater number of more realistic scenarios in ADAS/AD development and validation. It could also be applicable for other domains, e.g. in smart city planning, cooperative driving etc.

**Additional Feedback:**

No additional feedback

**Clarity:**

The paper is overall well readable, but seems to compact a lot of adjacent topics into one work. Sections vary in length (e.g. section 4.5)

**Correctness:**

The claims made in the paper are correct and the corresponding source code is provided.

**Documentation:**

The code is available, however the authors did not provide the full scenario database.

**Ethics:**

No concerns, simulated content only.

**Limitations:**

The authors have built a framework to load data from various sources. Unfortunately, this data (to my understanding) seems to only be usable in the metadrive simulator.

It would be great to also allow an export to open standards such as OpenScenario and OpenDrive. Due to the open source nature of this work, this might also be appendable by third parties.

**Opportunities For Improvement:**

The work does not explain in detail, how much closed-loop testing capability exists.
To my understanding, the behavior of ego and other participants is scripted given the existing real scenario. Therefore any interaction between ego and others, which does not conform to the pre-recorded scenario will necessitate a procedural change in vehicle behaviour (e.g. if ego slows down, vehicles behind need to do this as well). This is a challenging task and from the paper it is not obvious, if this capability exists and to what limitations.

Another minor open point is the usage of meta-information, e.g. time of day, weather, .. The demos show a night scene, but the paper does not explain how and what meta information is used.

**Relation To Prior Work:**

A small overlap exists to the previous work "Metadrive: Composing diverse driving scenarios for generalizable reinforcement learning", which describes the simulator and procedural map/scenario creation.

**Summary And Contributions:**

The authors present ScenarioNet, a toolchain to create scenario descriptions from multiple real-world sources along with a simulation framework.

The authors furthermore provide application examples in a comparison of different RL agents as well as an integration to ROS with the possibility to run OpenPilot.

---

> ### Author Response · Authors · 2023-08-09
> **Response to Reviewer DPk1**
>
> Thank you for your valuable time. We appreciate the suggestion of including more details regarding closed-loop simulation. Please find our elaboration on closed-loop simulation and responses to other minor concerns below.
>
> ---
>
> > **Q1: The work does not explain in detail, how much closed-loop testing capability exists.**
>
> **A1:** Actually, *ScenarioNet* already achieves closed-loop simulation by using the Intelligent Driver Model (lDM) policy. In *Section 3.3, Control Policies*, we mentioned that the log-replay policy can be replaced by the IDM policy to make the traffic reactive when the ego car deviates from the logged trajectory. To be specific, when initializing a scenario, vehicles behind the ego car will move along the trajectory with IDM policies rather than replaying the logged poses. The IDM policy can detect if there are objects present on its future trajectory and adjust the target velocity to avoid collision with the leading object. In this way, when the ego car shows behaviors different from the recorded ones, vehicles behind it can react to this change and show yielding behavior, which achieves closed-loop simulation.
>
> We provide a video to show this feature: https://drive.google.com/file/d/1Bif162uOwa3tX1uSWR_yNMDuOgBdFxYQ/view?usp=sharing. In this video, the red ego car with a yellow mark overhead is controlled manually with a keyboard, and other cars with purple and blue marks overhead are controlled by IDM policies and log-replay policies respectively. It shows that when we drive the ego car to new lanes or slow down, the cars with IDM policy will stop or decelerate to avoid collisions.
>
> IDM policy is widely used for closed-loop simulation and adopted by other platforms like Waymo Sim Agents[1] and nuPlan[2]. However, the IDM policy can not execute other complex behaviors like lane-changing, which makes the resulting behaviors limited and unrealistic. Thus a future direction is to replace the log-replayed policy and simple IDM policy with realistic and diverse traffic policies learned from the driving logs. We believe that *ScenarioNet* is a powerful tool to enable the exploration of this new direction.
>
> We will provide more details about the closed-loop simulation in the main paper and show the video on our website for highlighting this feature.
>
> [1] Montali, Nico, et al. "The waymo open sim agents challenge." arXiv preprint arXiv:2305.12032 (2023).
>
> [2] Caesar, Holger, et al. "nuplan: A closed-loop ml-based planning benchmark for autonomous vehicles." arXiv preprint arXiv:2106.11810 (2021).
>
>
> ---
>
>
>
>
> > **Q2: The full database is not provided.**
>
> **A2:** We provide all the data format converters for processing raw data of various public driving datasets from scratch so that the users can generate the full set by themselves. The main reason is to avoid violating the copyright and license of each dataset. For example, individual registration and license agreement are required for obtaining Waymo motion data and nuPlan/nuScenes data. Thus we can not release the full set directly to users without consent from the data provider. We plan to seek cooperation with the data provider so that users can access the converted data directly.
>
> ---
>
>
> > **Q3: The paper does not explain how and what meta information like time of day and whether is used**
>
> **A3:** The driving log doesn’t disclose the time of day and the weather. We just visualize this scenario to find the weather condition by eye-watching. In the future, this information can be extracted automatically with a visual prediction model.
>
> ---
>
> > **Q4: A small overlap exists to the previous work, MetaDrive**
>
> **A4:** The contribution of this work is a driving scenario management and simulation platform, which enables users to model diverse scenario data from heterogeneous datasets and simulate scenarios for machine learning applications. One of the underlying system modules, the simulation module, is built upon the MetaDrive simulator with substantial extensions on several aspects including:
>
> A unified scenario description including data structures, the definition of object types, tools for sanity-check, scenario summarization, and so on.
> A unified environment for closed-loop simulation, scenario database management, scenario reconstruction, and curriculum training.
> ROS bridge for AD stack testing
> Deferred rendering system for realistic 3D rendering
> CUDA and OpenGL interoperation for improving visual data collection efficiency
>
> We will discuss the difference in the revision.

---

### Official Review · Reviewer_7uZv · 2023-07-25
**Review of ScenarioNet**

**Rating:** 7
**Confidence:** 4

**Strengths:**

The paper's main strength lies in its approach to addressing the data accessibility issue in autonomous driving and the platform's ability to ingest diverse real world data sources. By using real-world traffic scenarios, the authors have been able to benchmark the decision-making of the AD system and improve the generalizability of the data-driven planning and control components. The paper also highlights the importance of aggregating as many scenarios as possible, as solely relying on driving data from one region or one database is insufficient to cover all potential traffic situations. The authors have also provided a ROS bridge for connecting open-sourced AD stacks and tested Openpilot, an end-to-end AD system.

There is definite strength to this submission and the underlying concept as it opens up the possibility of creating open-ended simulations out of static datasets, and to enhance the diversity of scenarios depicted in the datasets.

**Additional Feedback:**

None

**Clarity:**

The paper appears to be well-written, with clear explanations of the methodology and results. The authors have also provided a detailed discussion of the limitations and potential negative societal impacts of their work

**Correctness:**

The paper claims that the main claims made in the abstract and introduction accurately reflect the paper's contributions and scope. The authors have also included the code, data, and instructions needed to reproduce the main experimental results.

**Documentation:**

The associated code repository has some basic instructions for setting up the codebase and importing some of the supported datasets, merging datasets etc. But there is not detailed documentation on the API, and it seems quite hard to import custom datasets apart from the ones that are already being handled.

**Ethics:**

No ethics concerns.

**Limitations:**

The authors have discussed the limitations of their work in Section 5. They have also discussed potential negative societal impacts of their work in the same section.

**Opportunities For Improvement:**

The paper acknowledges that there are several critical issues that need to be addressed when building a large-scale data-driven simulation platform, and I believe this is a good first step but in a long journey. Specifically, ScenarioNet mainly focuses on replicating the scenarios and traffic behaviors (positions and dynamics of other vehicles, road structures, obstructions etc.); but not so much on the visual fidelity. Some of the sensor models might also not be as realistic, especially LiDAR which are quite hard to model in simulation and might require data-driven techniques. It might also be interesting to consider simulation engines such as Unreal Engine to allow for procedural generation of foliage etc to make the scenes more realistic.

Discussing how to import new datasets apart from the ones that were used in the paper in more detail could be very useful. The paper does mention "It is also convenient to write new converters for users’ own dataset..." - but what are the requirements of the datasets to be compatible with ScenarioNet? For example, not all datasets might have the level of detail contained in Waymo etc. in terms of ground truth 3D object positions, traffic light states etc. But in that case, it might be interesting to utilize state of the art segmentation/3D object pose models to extract pseudo-ground truth so as to leverage this powerful construct of creating a simulator out of unstructured data.

**Relation To Prior Work:**

The authors have cited the creators of the existing assets they used in their work. They have also provided a list of references, indicating a thorough review of the existing literature.

**Summary And Contributions:**

The paper presents ScenarioNet, a large-scale interactive scenario generation and benchmarking system for autonomous driving. The system is designed to address the data accessibility issue by collecting traffic scenarios from real-world driving datasets (Waymo, nuScenes etc.) and replaying them in simulation. This allows for the benchmarking of the decision-making of the autonomous driving (AD) system and improves the generalizability of the data-driven planning and control components. The authors have conducted a range of experiments based on ScenarioNet, including training a large scenario generation model, conducting cross-dataset single-agent training and testing experiments, and benchmarking Openpilot, an end-to-end AD stack in the reconstructed scenarios.

---

> ### Author Response · Authors · 2023-08-09
> **Response to Reviewer 7uZv**
>
> Thank you for your valuable time. We are glad to hear the constructive suggestions, which provide insightful directions for future development. Please find our detailed responses to your questions as follows.
>
> ---
>
> > **Q1: Visual fidelity could be improved in the future.**
>
> **A1**: We agree with this point. As *Reviewer Wnbm* is also interested in this issue, please find our detailed discussion in *Common Response,  How to improve visual fidelity*.
>
> ---
>
>
> > **Q2: It would be good to discuss how to import new datasets.**
>
> **A2:** There are some datasets that *ScenarioNet* doesn’t support at this stage. But one can easily convert them to the internal scenario description and build the database by following these steps:
>
> Download the original data and set up the official codebase for parsing this data. Usually, this step is irrelevant to *ScenarioNet*.
> Make a `convertor_function` which takes one scenario recorded in the original format as input and returns the scenario described in a new format. Actually, this process is like filling in the blank of the target scenario description by parsing the original data format with official APIs coupled with the new dataset.
> Scale up the data conversion with the built-in function, `write_to_directory`. It takes the `convert_function` and a list of scenario indices or original scenarios as input and writes the converted scenarios into a target directory. In this process, the multi-process parallel converting, sanity-check and other trivial steps will be finished automatically.
>
> Though step 2 seems time-consuming, most of the work in this step is, actually, calling APIs provided by the new dataset, retrieving the desired data, and putting them in the corresponding field. A good example is `convert_waymo_scenario`: https://github.com/metadriverse/scenarionet/blob/e9c1419a91c42e67fe176a538d2984cf1d9e0f93/scenarionet/converter/waymo/utils.py#L362 where most of the subfunction is called `extract_xyz`. Since the structure and fields to fill in the scenario description are clear and automatic tools are provided, we believe it would be easy for the community to add support for more datasets.
>
> We will elaborate on this process in the appendix. Moreover, we are going to provide technical guidance and examples in our documentation: https://scenarionet.readthedocs.io/en/latest/ for providing detailed technical guidance.
>
>
> ---
>
> > **Q3: Lack of API reference and documentation.**
>
> **A3:** We have built documentation at: https://scenarionet.readthedocs.io/en/latest/ The detailed plan can be found at *Common Response, Documentation*.

---

### Author Response · Authors · 2023-08-09
**Common Response**

# Common Response

We thank all reviewers for their insightful reviews. We revise our submission with modified parts highlighted in red, and summarize the major updates and some common concerns as follows:

---

### New Experiment

As suggested by *Reviewer KCPD*, we conduct a new *nuplan+pg w/ curriculum* experiment. It suggests that aggregating synthetic scenarios to the training data can benefit policy learning and improve the *Success Rate* when evaluating agents in real-world scenarios. This is because as shown in Figure 4, almost every synthetic scenario contains bends, which only exist in approximately 20% of real-world scenarios, and thus the added synthetic scenarios actually balance the training dataset. As a result, agents trained on this more diverse combined dataset acquire a better ability to follow a curved trajectory, inducing a lower *Out of Road Rate*. We have included this new result and discussion in our paper.

---

### How to improve visual fidelity

As suggested by *Reviewer 7U7V* and *Reviewer Wnbm*, the visual fidelity of the closed-loop simulation can be further improved with the collected raw sensor data. We already have some plans to improve it and will include this discussion in the paper for sharing our thoughts regarding realistic sensor simulation.

Currently, the sensor simulation including the camera and lidar is achieved in a Computer Graphics (CG) way, where people try restoring the mesh and texture for objects from real-world data and shading these models based on lights and materials to make them visually realistic. To improve the rendering results of *ScenarioNet*, we do plan to reconstruct the geometry data for traffic participants and objects from the driving videos. Besides, we also consider connecting *ScenarioNet* with Unreal Engine or Nvidia Omniverse in the future as they could provide better shading results.

On the other hand, NeRF is an alternative to improve the quality of sensor simulation. Through volume rendering, it can directly synthesize new camera views and point clouds from the driving videos when traffic participants and the ego car move with different trajectories and poses in the closed-loop training. This way is purely data-driven and can exempt the need for restoring 3D assets like objects and buildings. Recent results [1][2][3] already demonstrate its potential in terms of camera and lidar simulation. However, how to make the NeRF scene editable is still an open problem; hence, we plan to investigate this in the future.

---

### Technical Limitations
As suggested by Reviewer Ei6A, we will include more discussion on the technical limitation of *ScenarioNet*. We believe this can help researchers figure out if ScenarioNet is suitable for their aims.

Two potential technical drawbacks exist in *ScenarioNet*. One drawback is that the simulator runs on CPU, while some latest simulators like Isaac-Gym support GPU-based parallel simulation and thus outperform *ScenarioNet* in terms of sample efficiency. A solution is to collect data in parallel environments with tools like EnvPool. Another drawback is that *ScenarioNet* doesn’t support differential simulation which allows training policy with the derivative of reward function directly. In the future, *ScenarioNet* can address this by introducing a simple bicycle model for vehicle dynamics simulation, making the system differentiable. We will include the discussion above in the new version of the paper.

---

### Documentation

As suggested by *Reviewer 7U7V* and *Reviewer Ei6A*, it would be good to have documentation to explain system design, API reference, and how to support third-party contribution and customization. Therefore, we have built documentation at:https://scenarionet.readthedocs.io/en/latest/. In the rebuttal period, we will provide details about the internal scenario description, API reference, system design, and guidelines for third-party contribution. Also, illustrative examples will be included such as importing new datasets, running simulations, and training models.

---

### Scenario Gallery

We made a gallery to visualize the real-world scenarios at https://metadriverse.github.io/snetasset/snet-asset.html. We believe it is more informative than the figures shown in the Appendix. Feel free to browse the gallery, and we hope you enjoy this work!

---

### Reference

[1] Yang, Ze, et al. "UniSim: A Neural Closed-Loop Sensor Simulator." Proceedings of the IEEE/CVF Conference on Computer Vision and Pattern Recognition. 2023.

[2] Xie, Ziyang, et al. "S-NeRF: Neural Radiance Fields for Street Views." The Eleventh International Conference on Learning Representations. 2022.

[3] Wu, Zirui, et al. "MARS: An Instance-aware, Modular and Realistic Simulator for Autonomous Driving." arXiv preprint arXiv:2307.15058 (2023).

---

> ### Author Response · Authors · 2023-08-28
> **Any post-rebutall suggestions?**
>
> Dear Reviewers,
>
> Please kindly let us know if you have any post-rebuttal feedback. If something is still unclear, we would be happy to provide alternative explanations or clarifications on any other part of the paper.
>
> Best,
>
> Authors

---

### Decision · Program_Chairs · 2023-09-22

**Decision:**

Accept (Poster)

**Comment:**

Significance of Contributions: As clearly articulated by the reviews, the strength of the work is the development of a large-scale interactive scenario generation and benchmarking system for autonomous driving. The system provides enhanced data accessibility by allowing the collection traffic scenarios from real-world driving datasets (Waymo, nuScenes etc.), replaying them in simulation and allowing benchmarking of decision making in autonomous driving settings. A range of experiments include:
training a large scenario generation model, conducting cross-dataset single-agent training and testing experiments, and benchmarking Openpilot, an end-to-end AD stack in the reconstructed scenarios.

Strengths:  Enhanced data accessibility and sampling from diverse real-world traffic scenarios
Benchmarking for generalizing data driven planning and control in autonomous driving
ROS bridge for connecting open-sourced AD stacks and tested Openpilot, an end-to-end AD system.
Possibility of creating open-ended simulations out of static datasets, and to enhance the diversity of scenarios depicted in the datasets.

Limitations: The authors have discussed the limitations of their work and they have also discussed potential negative societal
impacts of their work in the same section.

The rebuttal and discussions that follow are convincing and the reviewers are overall positive about acceptance of the paper.

Recommendation:  I recommend the paper for a poster. It may be appropriate also for a spotlight if the relative ranking of this paper is high compared to papers accepted.